# Conversion of random X-inactivation to imprinted X-inactivation by maternal PRC2

**Clair Harris[1][†], Marissa Cloutier[1][†], Megan Trotter[1], Michael Hinten[1][‡], Srimonta Gayen[1][§], Zhenhai Du[2], Wei Xie[2], Sundeep Kalantry[1]\***

[1]Department of Human Genetics, University of Michigan, Ann Arbor, United States; [2]Center for Stem Cell Biology and Regenerative Medicine, MOE Key Laboratory of Bioinformatics, School of Life Sciences, THU-PKU Center for Life Science, Tsinghua University, Beijing, China

**Abstract** Imprinted X-inactivation silences genes exclusively on the paternally-inherited X-chromosome and is a paradigm of transgenerational epigenetic inheritance in mammals. Here, we test the role of maternal vs. zygotic Polycomb repressive complex 2 (PRC2) protein EED in orchestrating imprinted X-inactivation in mouse embryos. In maternal-null ($Eed^{m-/-}$) but not zygotic-null ($Eed^{-/-}$) early embryos, the maternal X-chromosome ectopically induced $Xist$ and underwent inactivation. $Eed^{m-/-}$ females subsequently stochastically silenced $Xist$ from one of the two X-chromosomes and displayed random X-inactivation. This effect was exacerbated in embryos lacking both maternal and zygotic EED ($Eed^{mz-/-}$), suggesting that zygotic EED can also contribute to the onset of imprinted X-inactivation. $Xist$ expression dynamics in $Eed^{m-/-}$ embryos resemble that of early human embryos, which lack oocyte-derived maternal PRC2 and only undergo random X-inactivation. Thus, expression of PRC2 in the oocyte and transmission of the gene products to the embryo may dictate the occurrence of imprinted X-inactivation in mammals.
DOI: https://doi.org/10.7554/eLife.44258.001

**\*For correspondence:**
kalantry@umich.edu

[†]These authors contributed equally to this work

**Present address:** [‡]Division of Nephrology and Hypertension, Mayo Clinic College of Medicine, Rochester, United States; [§]Department of Molecular Reproduction, Development and Genetics, Indian Institute of Science, Bangalore, India

**Competing interests:** The authors declare that no competing interests exist.

## Introduction

X-chromosome inactivation results in the mitotically-stable transcriptional inactivation of one of the two X-chromosomes in female mammals in order to equalize X-linked gene expression between males and females (*Morey and Avner, 2011*; *Plath et al., 2002*). Two different forms of X-inactivation characterize the mouse embryo, imprinted and random. Imprinted X-inactivation results in the silencing of genes exclusively on the paternal X-chromosome and initiates during preimplantation embryogenesis (*Huynh and Lee, 2003*; *Mak et al., 2004*; *Monk and Kathuria, 1977*; *Okamoto et al., 2004*; *Takagi and Sasaki, 1975*). In later stage embryos, imprinted X-inactivation of the paternal-X is stably maintained in the extraembryonic lineage but reversed in the embryonic lineage (*Harper et al., 1982*; *Mak et al., 2004*; *Okamoto et al., 2004*; *Takagi and Sasaki, 1975*; *West et al., 1977*), which subsequently undergoes random inactivation of either the maternal or the paternal X-chromosome (*Lyon, 1961*). Notably, imprinted X-inactivation is a paradigm for both mitotic as well as meiotic, or transgenerational, epigenetic regulation, due to its stable parent-of-origin-specific inactivation pattern.

X-inactivation is characterized by a well-defined series of epigenetic events (*Kalantry, 2011*). Both imprinted and random X-inactivation are prefaced by the expression of X-linked non-protein coding Xist RNA from the prospective inactive-X (*Kay et al., 1994*; *Penny et al., 1996*). During imprinted X-inactivation in the mouse embryo, $Xist$ is expressed at the two-cell stage and the RNA visibly begins to coat the paternal-X at the four-cell stage (*Kalantry et al., 2009*; *Namekawa et al.,*

**eLife digest** Almost every one of our cells, with a few exceptions, contains the complete set of genes needed to build and maintain the human body. Yet, not all of these genes are active in every cell. Instead, some genes are tagged for activation, while others are silenced. These changes do not alter the genetic code, only how it is read by the cell, and are collectively referred to as epigenetics.

Female mammals have two X-chromosomes compared to males' one. As such, females will silence one of those chromosomes to avoid getting a double-dose from those genes located on the X-chromosome. This epigenetic process is called X-chromosome inactivation, and it lasts for the life of the animal.

Epigenetic information can also be passed on to future generations. In early female embryos of mice, for example, it is always the X-chromosome inherited from the father that is suppressed, which suggests that the instructions for which X-chromosome to inactivate must have come from the parents.

Harris, Cloutier et al. set out to dissect the mechanics of the specialised form of X-chromosome inactivation seen in female embryos of mice, which is known as imprinted X-inactivation. A protein called EED was suspected to play a key role. Embryos inherit EED protein from the mother's egg, so it was reasoned that this protein may be the epigenetic link between the generations. The cascade of epigenetic events leading to imprinted X-inactivation in the early embryo has been well-defined, but the role of maternal EED was yet to be tested.

The experiments showed that the mother's EED protein was needed to silence the father's X-chromosome in female mouse embryos. Without EED from the mother's egg, early embryos failed to initiate imprinted X-inactivation and reverted instead to random X-inactivation, where either X-chromosome is chosen for silencing in female cells. This pattern resembles what happens in early human embryos, which are unable to undergo imprinted X-inactivation because a woman's eggs lack the EED protein.

Together these new findings trace the passage of epigenetic information from parent to offspring at the molecular level. With evidence like this, scientists can better understand mechanisms of non-genetic inheritance more broadly, including from parent to offspring.
DOI: https://doi.org/10.7554/eLife.44258.002

2010; *Patrat et al., 2009*). The progressive accumulation of Xist RNA coincides with the gradual and stereotyped silencing of paternal X-linked genes that is only completed after the blastocyst stage of embryogenesis (*Kalantry et al., 2009*; *Namekawa et al., 2010*; *Patrat et al., 2009*). Coincident with Xist RNA coating, Polycomb repressive complex 2 (PRC2) proteins and the PRC2-catalyzed chromatin mark histone H3K27me3 accumulate on the inactive-X, correlating with the silencing of X-linked genes (*Mak et al., 2004*; *Okamoto et al., 2004*; *Plath et al., 2003*; *Silva et al., 2003*). Moreover, the mis-expression of *Xist* results in the concomitant accumulation of PRC2 proteins and H3K27me3 (*de la Cruz et al., 2005*; *Kohlmaier et al., 2004*; *Plath et al., 2003*; *Silva et al., 2003*), suggesting that Xist RNA directly or indirectly recruits PRC2 to the inactive-X. PRC2 has thus been suggested to contribute to the establishment of X-inactivation (*Plath et al., 2003*; *Silva et al., 2003*).

Consistent with a role for PRC2 in X-inactivation, we and others previously showed that post-implantation female mouse embryos mutant for the Polycomb gene *Eed* fail to maintain silencing of paternal X-linked genes during imprinted X-inactivation (*Kalantry and Magnuson, 2006*; *Kalantry et al., 2006*; *Wang et al., 2001*). EED is a non-catalytic component of the PRC2 complex, but EED binding to the PRC2 enzyme EZH2 is required for the full methyltransferase activity of EZH2 (*Cao et al., 2002*; *Czermin et al., 2002*; *Kuzmichev et al., 2002*; *Müller et al., 2002*). When EED is mutated other core PRC2 proteins are degraded and the histone H3K27me3 mark is lost (*Montgomery et al., 2005*). Thus, EED is an essential component of PRC2 and EED function is canonically equated with H3K27me3 catalysis (*Margueron and Reinberg, 2011*; *Montgomery et al., 2005*).

Although *Eed*$^{-/-}$ embryos fail to maintain imprinted X-inactivation, the mutant embryos initiate imprinted X-inactivation properly (*Kalantry and Magnuson, 2006*; *Kalantry et al., 2006*). A

potential answer for this difference is that *Eed*[-/-] embryos inherit maternal EED protein that is present in the oocyte (*Kalantry and Magnuson, 2006*; *Plath et al., 2003*; *Shumacher et al., 1996*). The presence of maternally-derived EED protein could explain the absence of a defect in establishing imprinted X-inactivation in *Eed*[-/-] embryos. Such maternal control of imprinted X-inactivation would also be consistent with a transgenerational epigenetic effect that underlies genomic imprinting (*Barlow, 2011*; *Ferguson-Smith and Bourc'his, 2018*; *Lee and Bartolomei, 2013*; *van Otterdijk and Michels, 2016*). Here, we test the hypothesis that oocyte-derived PRC2 orchestrates imprinted X-inactivation in the early embryo.

## Results

### EED and H3K27me3 enrichment on the inactive-X in *Eed*[-/-] embryos

PRC2 proteins and H3K27me3 are first enriched on the prospective inactive paternal X-chromosome in the early mouse embryo at the 8–16 cell morula stage (*Okamoto et al., 2004*). We assessed the accumulation of EED, H3K27me3, and Xist RNA by immunofluorescence (IF) combined with RNA fluorescent *in situ* hybridization (FISH) in wild-type (WT) embryonic day (E) 3.5 blastocyst embryos (*Cloutier et al., 2018*; *Hinten et al., 2016*), which are in the process of silencing paternal X-linked genes and establishing imprinted X-inactivation (*Borensztein et al., 2017*; *Namekawa et al., 2010*; *Patrat et al., 2009*; *Wang et al., 2016*). As expected, females displayed coincident accumulation of EED, H3K27me3, and Xist RNA in a vast majority of the nuclei (72–100%). Males, by contrast, lacked such enrichment (*Figure 1A*).

Our previous work suggested that zygotically-null preimplantation embryos harbor WT maternal EED protein (*Kalantry and Magnuson, 2006*; *Kalantry et al., 2006*). To test for the presence of maternally-derived EED protein in *Eed*[-/-] embryos, we employed our previously generated conditional *Eed* mutation (*Figure 1—figure supplement 1A*) (*Maclary et al., 2017*). We generated E3.0-E3.5 blastocyst-stage embryos zygotically-null and heterozygous for *Eed* (*Eed*[-/-] and *Eed*[+/-], respectively) from a cross of *Eed*[+/-] females with *Eed*[fl/-];*Prm-Cre* males. *Prm-Cre* is active during spermatogenesis and catalyzes the deletion of the *lox*p flanked (floxed) *Eed* allele in the mature sperm (*Figure 1—figure supplement 1B*) (*O'Gorman et al., 1997*). As a result, about half of the embryos generated from the above cross are expected to be genotypically *Eed*[-/-] and the other half *Eed*[+/-]. In the derived embryos, we assayed inactive-X enrichment of EED, H3K27me3, and Xist RNA by combined IF/FISH (*Figure 1B*). Of the 41 female embryos examined, nine showed coincident accumulation of EED and/or H3K27me3 with Xist RNA in over 70% of the nuclei and were not significantly different from WT embryos in *Figure 1A* (p>0.1). An additional nine embryos were devoid of EED or H3K27me3 enrichment overlapping with the Xist RNA coat. We presumed the former to be *Eed*[+/-] embryos and the latter to be *Eed*[-/-] embryos. The remaining 23 embryos displayed 2–70% of nuclei with EED and/or H3K27me3 enrichment. This intermediate class likely represents *Eed*[+/-] or *Eed*[-/-] embryos that had not yet fully depleted maternally-inherited EED protein or *Eed*[+/-] embryos which had not yet robustly expressed zygotic EED. Male embryos from the cross, distinguished by a lack of Xist RNA coating, did not show enrichment of EED or H3K27me3 in the nucleus, as in the WT male embryos in *Figure 1A*.

To confirm that there is no bias in the sex ratio or genotype of the embryos, we performed PCR genotyping of embryos derived from the above cross (*Figure 1C*). Embryos from 12 litters showed no statistical difference in the distribution of *Eed*[+/-] and *Eed*[-/-] male or female embryos (p>0.05), suggesting that the intermediate class of 23 embryos in *Figure 1A* are likely a mixture of *Eed*[+/-] or *Eed*[-/-] embryos. Together, the results in *Figure 1* suggest that genotypically null *Eed*[-/-] embryos inherit oocyte-derived maternal EED protein and that expression of EED transitions from maternal to zygotic at or slightly before the blastocyst stage.

To define the kinetics of depletion of maternal EED and induction of zygotic EED prior to the blastocyst stage, we quantified EED and H3K27me3 nuclear IF signals in 2-, 4-, 8-, and 16 cell embryos from the following series of crosses. The first was *Eed*[fl/fl] females crossed to *Eed*[fl/fl] males, which yielded control *Eed*[fl/fl] embryos. The second was a cross of *Eed*[fl/-] females to *Eed*[fl/-];*Prm-Cre* males to generate *Eed*[fl/-] and *Eed*[-/-] embryos (*Eed*[fl/-] / *Eed*[-/-]). Whereas both *Eed*[fl/-] and *Eed*[-/-] embryos are expected to harbor maternal EED protein, *Eed*[fl/-] but not *Eed*[-/-] embryos would express zygotic EED. The third cross was of *Eed*[fl/fl];*Zp3-Cre* females to WT males to yield embryos that are

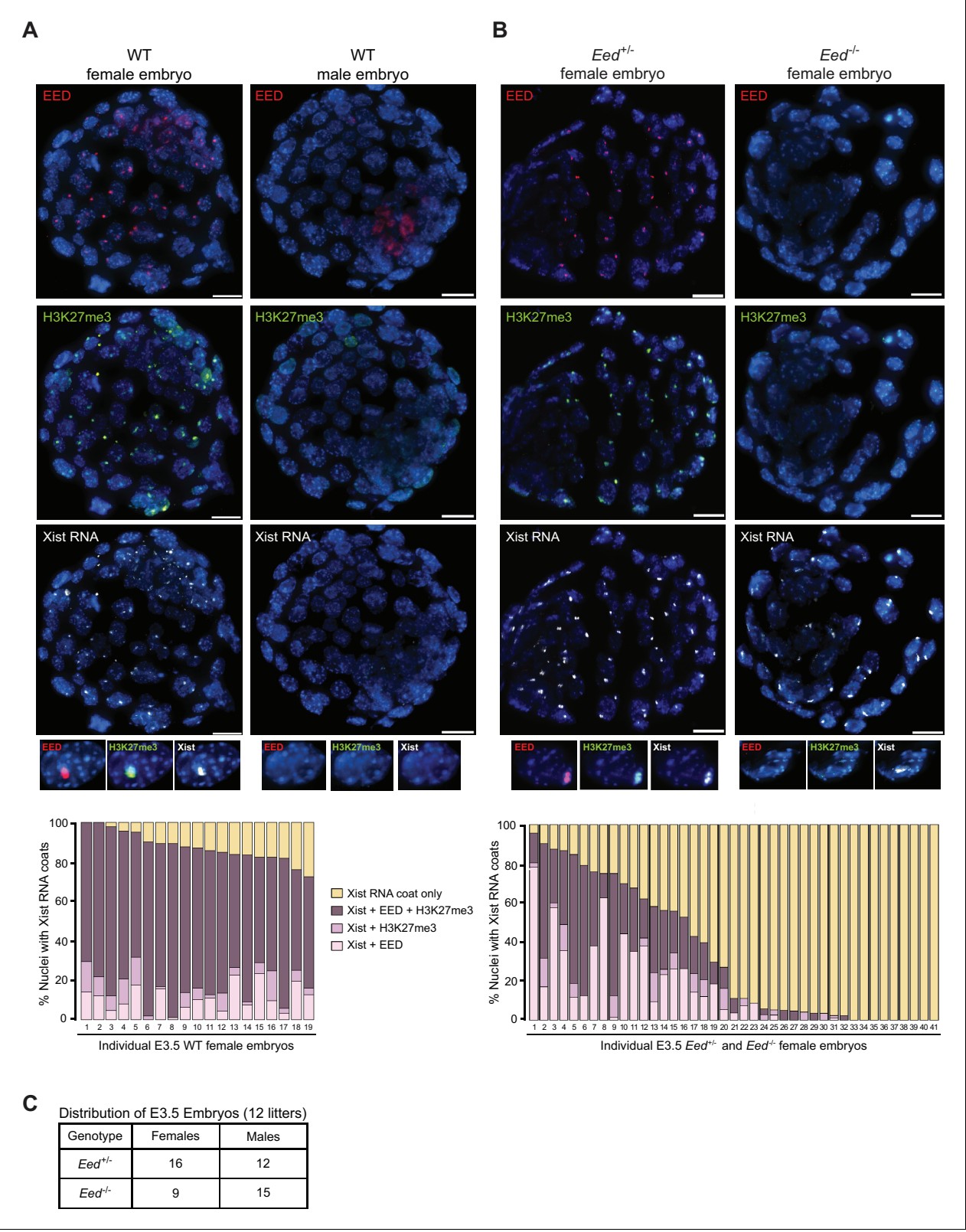

**Figure 1.** Coincident accumulation of EED and H3K27me3 on the inactive X-chromosome in blastocyst-stage WT, *Eed⁺/⁻* and *Eed⁻/⁻* mouse embryos. See also *Figure 1—figure supplement 1*. (A,B) RNA FISH detection of Xist RNA (white) and immunofluorescence (IF) detection of EED (red) and H3K27me3 (green) in representative female and male wild-type (WT) (A) or female *Eed⁺/⁻* and *Eed⁻/⁻* (B) E3.0 – E3.5 blastocyst embryos. Nuclei are stained blue with DAPI. Scale bars, 20 μm. Embryos ranged in size from 23 to 57 nuclei. Bar plots, percentage of nuclei with coincident accumulation of

*Figure 1 continued on next page*

**Figure 1 continued**

Xist RNA and EED and/or H3K27me3 enrichment in individual embryos. (C) Genotype and sex distribution of $Eed^{+/-}$ and $Eed^{-/-}$ mouse blastocyst embryos from the cross in (B). The difference between the frequency of $Eed^{+/-}$ vs $Eed^{-/-}$ male and female embryos is not significant (p>0.05, Two-tailed Student's T-test).

DOI: https://doi.org/10.7554/eLife.44258.003

The following figure supplement is available for figure 1:

**Figure supplement 1.** Generation of $Eed^{-/-}$ embryos.

DOI: https://doi.org/10.7554/eLife.44258.004

devoid of maternal EED ($Eed^{m-/-}$) but which are capable of expressing zygotic EED. $Zp3$-$Cre$ is active in the growing oocyte, where it efficiently deletes the $Eed^{fl}$ allele and generates embryos devoid of maternal EED (Figure 5C and *Figure 2—figure supplement 1A*) (*Lewandoski et al., 1997*). The final cross was a cross of $Eed^{fl/fl}$;$Zp3$-$Cre$ females with $Eed^{fl/fl}$;$Prm$-$Cre$ males to generate embryos devoid of both maternal and zygotic EED ($Eed^{mz-/-}$).

$Eed^{fl/fl}$ and $Eed^{fl/-}$ / $Eed^{-/-}$ 2-cell embryos exhibited similar levels of EED and H3K27me3, whereas $Eed^{m-/-}$ and $Eed^{mz-/-}$ embryos were devoid of both EED and H3K27me3 (*Figures 2A, C and D*; *Supplementary file 1*). These data are consistent with the 2-cell embryo harboring only maternally-derived EED and H3K27me3. Four-cell embryos displayed a similar pattern to 2-cell embryos, although a subset of $Eed^{fl/-}$ / $Eed^{-/-}$ ~4-cell embryos displayed reduced EED and H3K27me3 levels, consistent with expression of zygotic EED beginning at or slightly before this stage and its failure in $Eed^{-/-}$ embryos (*Figure 2C* and *Figure 2—figure supplement 1B*; *Supplementary file 1*). At the ~8-cell stage, $Eed^{fl/-}$ / $Eed^{-/-}$ embryos showed highly variable EED and H3K27me3 levels, suggesting further differentiation of the two genotypes. In agreement with increasing zygotic $Eed$ expression, $Eed^{m-/-}$ ~8-cell embryos displayed higher levels of EED and H3K27me3 than the corresponding $Eed^{mz-/-}$ embryos (*Figure 2C* and *Figure 2—figure supplement 1B*; *Supplementary file 1*). By the ~16-cell stage, $Eed^{fl/-}$ / $Eed^{-/-}$ embryos were clearly separated into two categories. One group had statistically lower levels of EED, while the other group was statistically indistinguishable from $Eed^{fl/fl}$ embryos (*Figures 2B, C and D*; *Supplementary file 1*). Therefore, the likely genotypes of the two groups are $Eed^{-/-}$ and $Eed^{fl/-}$, respectively. $Eed^{m-/-}$ 16-cell embryos continued to display higher levels of EED and H3K27me3 than the $Eed^{mz-/-}$ embryos, but nevertheless harbored significantly lower EED and H3K27me3 levels than $Eed^{fl/fl}$ embryos (*Figures 2B, C and D*; *Supplementary file 1*). In order to visualize how EED levels are changing across early embryogenesis, we plotted the mean fluorescence intensity values of EED for each genotype by embryonic stage (*Figure 2E*). Maternally-derived EED starts declining at ~4-cell stage but is still present at the 16-cell stage. Conversely, while zygotic $Eed$ transcription initiates at ~4-cell stage, zygotic EED levels are still low in ~16-cell embryos, suggesting that EED in WT $Eed^{fl/fl}$ 16-cell embryos is a combination of maternally-derived and zygotically generated protein (*Figure 2F*).

## Imprinted X-inactivation initiation in $Eed^{-/-}$ embryos

To test if zygotic $Eed^{-/-}$ embryos initiate and establish imprinted X-inactivation of the paternal X-chromosome, we compared X-linked gene expression in an allele-specific manner in individual hybrid $Eed^{fl/fl}$, $Eed^{fl/-}$, and $Eed^{-/-}$ E3.5 blastocysts by RNA sequencing (RNA-Seq) (*Figure 3—figure supplement 1A*). In these embryos, the maternal X chromosome was derived from the *Mus musculus* 129/S1 mouse strain and the paternal-X from the divergent *Mus molossus* JF1/Ms strain (Materials and methods). We exploited single nucleotide polymorphisms (SNPs) to assign RNA-Seq reads to either the maternal or paternal X-chromosome in the hybrid embryos (*Cloutier et al., 2018*; *Maclary et al., 2017*). A subset of X-linked genes was expressed more robustly from the paternal allele relative to the maternal allele in $Eed^{fl/-}$ and $Eed^{-/-}$ female embryos compared to $Eed^{fl/fl}$ embryos (*Figure 3A*; *Supplementary file 2*). However, when the allelic expression ratio of all X-linked genes in *Figure 3A* was averaged, paternal X-linked gene expression was not significantly higher in $Eed^{-/-}$ blastocysts compared to $Eed^{fl/-}$ (p = 0.72) or $Eed^{fl/fl}$ (p = 0.76) female embryos (*Figure 3B* and *Figure 3—figure supplement 1B*; *Supplementary file 2* and *Supplementary file 3*). X-linked genes were expressed predominantly from the maternal allele in all three genotypes. Thus,

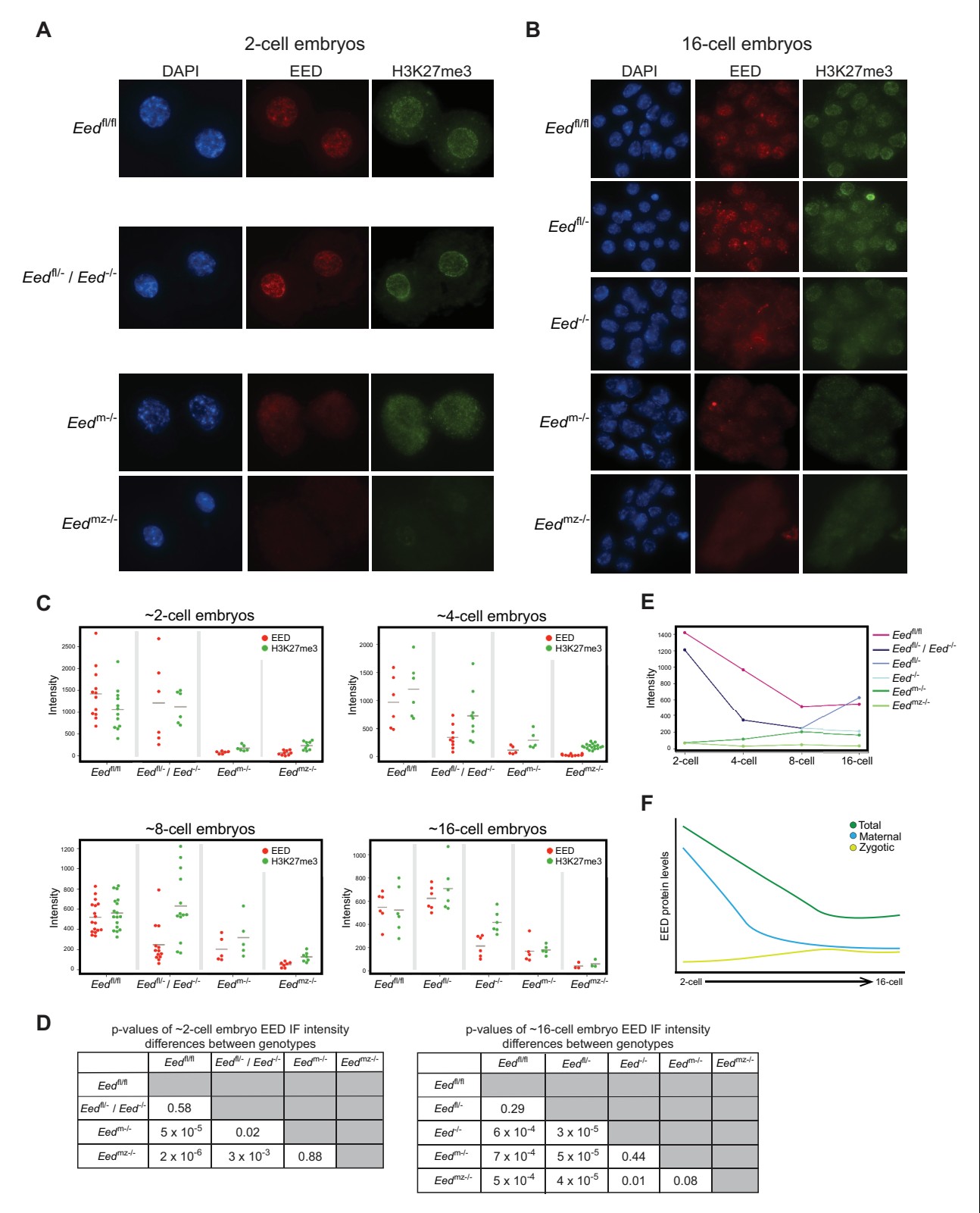

**Figure 2.** Assessment of maternal and zygotic EED expression in early preimplantation embryos. See also *Figure 2—figure supplement 1*, and *Figure 2—source data 1*. (A,B) Immunofluorescent (IF) detection of EED (red) and H3K27me3 (green) in 2- and 16-cell *Eed*$^{fl/fl}$, *Eed*$^{fl/-}$ / *Eed*$^{-/-}$, *Eed*$^{m-/-}$, and *Eed*$^{mz-/-}$ embryos. Nuclei are stained blue by DAPI. (C) Dot plots of EED and H3K27me3 IF signals in the five genotypes (*Eed*$^{fl/fl}$, *Eed*$^{fl/-}$, *Eed*$^{-/-}$, *Eed*$^{m-/-}$, *Eed*$^{mz-/-}$) at the ~2-cell,~4-cell, ~8-cell, and ~16-cell stage. Each dot represents an individual embryo. The gray line indicates mean fluorescence

*Figure 2 continued on next page*

*Figure 2 continued*

intensity. Pairwise statistical comparisons between all genotypes are included in **Supplementary file 1**. (D) Significance testing of differences in EED fluorescence intensity in ~2-cell embryos and ~16-cell embryos plotted in (**C**) (Two-tailed Student's T-test). (E) Mean EED fluorescence intensity from data in (**C**) plotted across early embryogenesis. (F) Model of change in maternal, zygotic, and total EED expression levels during early embryonic development.

DOI: https://doi.org/10.7554/eLife.44258.005

The following source data and figure supplement are available for figure 2:

**Source data 1.** Raw IF intensity data of individual nuclei.
DOI: https://doi.org/10.7554/eLife.44258.007
**Figure supplement 1.** Analysis of EED and H3K27me3 fluorescence intensity in *Eed* mutants.
DOI: https://doi.org/10.7554/eLife.44258.006

the ratio of maternal:paternal X-linked gene expression in $Eed^{-/-}$ female blastocysts was broadly similar to that in $Eed^{fl/fl}$ and $Eed^{fl/-}$ embryos.

We next sought to validate the RNA-Seq data via Pyrosequencing. Pyrosequencing is a low-throughput technique that can accurately capture allelic expression ratios of individual genes (**Cloutier et al., 2018**; **Gayen et al., 2015**). We analyzed the expression of *Xist* and three X-linked genes subject to X-inactivation, *Rnf12*, *Atrx*, and *Pgk1*. *Xist* expression analysis by Pyrosequencing was especially important, as there was variability in *Xist* SNP-overlapping read coverage in the RNA-Seq data due potentially to the highly repetitive sequence of Xist RNA. We did not detect any significant changes in maternal:paternal allelic expression in hybrid $Eed^{-/-}$ vs. $Eed^{fl/fl}$ and $Eed^{fl/-}$ blastocysts (**Figure 3C** and **Figure 3—figure supplement 1C**; **Supplementary file 4**). Whereas *Xist* was expressed predominantly from the paternal allele, *Rnf12*, *Atrx*, and *Pgk1* were preferentially expressed from the maternal allele in all three genotypes.

As an independent validation of the RNA-Seq and Pyrosequencing results, we also performed RNA FISH to test Xist RNA coating and nascent RNA expression of *Rnf12* in $Eed^{-/-}$ and $Eed^{fl/fl}$ female (**Figure 3D**) and male (**Figure 3—figure supplement 1D**) blastocysts. RNA FISH has the added benefit of providing single cell expression resolution in embryos (**Cloutier et al., 2018**; **Hinten et al., 2016**). We distinguished $Eed^{fl/fl}$ from $Eed^{-/-}$ female embryos by assaying H3K27me3 enrichment by IF on the Xist RNA-coated X-chromosome (**Figure 3D and E**). We classified embryos displaying fewer than 5% of the nuclei with this H3K27me3 enrichment as $Eed^{-/-}$ (**Figure 3E**). Xist RNA coating and *Rnf12* expression in female $Eed^{-/-}$ embryos did not differ significantly from $Eed^{fl/fl}$ blastocysts (**Figure 3D and F**). Both sets of embryos displayed Xist RNA coating of one X-chromosome and *Rnf12* expression from the other X-chromosome in a majority of the cells. Male $Eed^{-/-}$ or $Eed^{+/-}$ embryos also did not differ significantly from $Eed^{fl/fl}$ embryos in their *Rnf12* expression patterns (**Figure 3—figure supplement 1D**). Thus, by three independent assays – allele-specific RNA-Seq, Pyrosequencing, and RNA FISH – zygotic *Eed* expression appears to be largely dispensable for the initiation and establishment of imprinted X-inactivation.

## Defective imprinted X-inactivation initiation in $Eed^{m-/-}$ embryos

Since early $Eed^{-/-}$ embryos harbor WT oocyte-derived EED protein, we next examined the role of maternal EED in initiating imprinted X-inactivation in $Eed^{m-/-}$ and $Eed^{mz-/-}$ blastocysts, which are devoid of oocyte-derived EED. $Eed^{m-/-}$ blastocysts exhibited a small percentage of nuclei with H3K27me3 enrichment coinciding with the Xist RNA coat (**Figure 4A**). $Eed^{mz-/-}$ blastocysts, on the other hand, lacked all such overlapping accumulation (**Figure 4A**). H3K27me3 enrichment on the Xist RNA-coated X-chromosome in $Eed^{m-/-}$ but not $Eed^{mz-/-}$ blastocysts is likely due to the expression of zygotic *Eed* in $Eed^{m-/-}$ but not $Eed^{mz-/-}$ embryos (**Figure 2**).

To test if maternal EED regulates imprinted X-inactivation, we conducted allele-specific RNA-Seq on individual hybrid $Eed^{m-/-}$ and $Eed^{mz-/-}$ E3.5 blastocysts (**Figure 4—figure supplement 1A**). Strikingly, the RNA-Seq data revealed a relative increase in paternal X-linked gene expression in $Eed^{m-/-}$ and $Eed^{mz-/-}$ embryos compared to $Eed^{fl/fl}$, $Eed^{fl/-}$, and $Eed^{-/-}$ embryos (**Figure 4B and C**, and **Figure 4—figure supplement 1B**; **Supplementary file 2** and **Supplementary file 3**). Furthermore, $Eed^{mz-/-}$ embryos appeared to express paternal X-linked genes to a greater degree compared to $Eed^{m-/-}$ embryos (**Figure 4B**). When allelic expression ratios of all X-linked genes in **Figure 4B** were

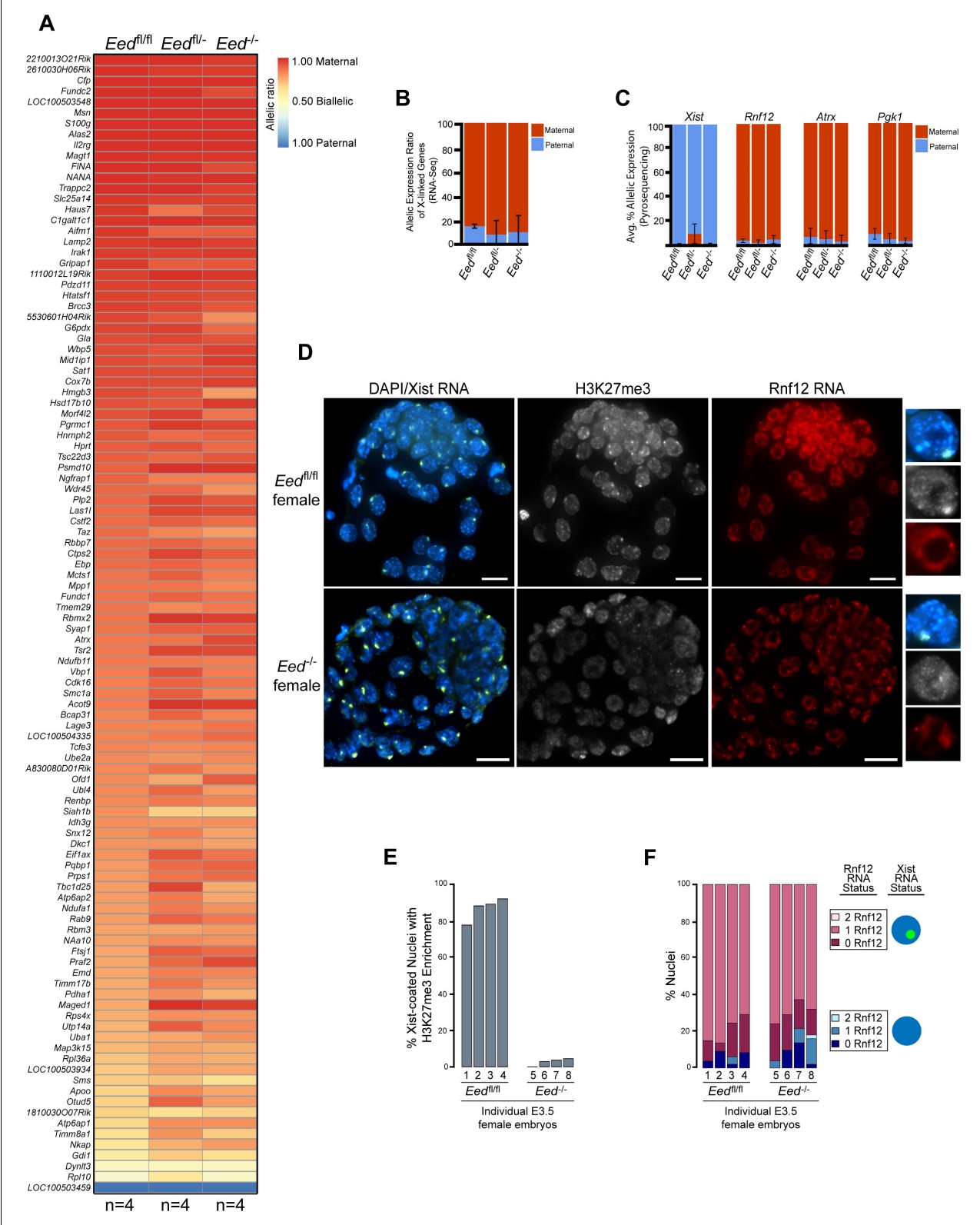

**Figure 3.** Lack of defective X-inactivation initiation in *Eed*[-/-] blastocysts. See also *Figure 3—figure supplement 1*. (**A**) Allele-specific X-linked gene expression heat map of female *Eed*[fl/fl], *Eed*[fl/-], and *Eed*[-/-] blastocysts. Four embryos each of *Eed*[fl/fl], *Eed*[fl/-], and *Eed*[-/-] genotypes were sequenced individually and only genes with informative allelic expression in all samples are plotted (see Materials and methods). Genes are ordered on the basis of allelic expression in *Eed*[fl/fl] embryos. (**B**) Average allelic expression of the RNA-Seq data shown in (**A**). The mean allelic expression of X-linked genes

*Figure 3 continued on next page*

*Figure 3 continued*
lacks significant difference between each combination of the three genotypes (p>0.05, Welch's two-sample T-test). Pairwise statistical comparisons between all genotypes are included in **Supplementary file 3**. (C) Pyrosequencing-based quantification of allelic expression of X-linked genes *Xist*, *Rnf12*, *Atrx* and *Pgk1* in *Eed*^fl/fl, *Eed*^fl/-, and *Eed*^-/- blastocysts. Error bars represent the standard deviation of data from 3 to 6 independent blastocyst embryos. The mean allelic expression of all four genes lack significant difference between each combination of the three genotypes (p>0.05, Welch's two-sample T-test). Pairwise statistical comparisons for all genes and between all genotypes are included in **Supplementary file 4**. (D) RNA FISH detection of Xist RNA (green), Rnf12 RNA (red), and IF detection of H3K27me3 (white) in representative *Eed*^fl/fl or *Eed*^-/- female blastocysts. Nuclei are stained blue with DAPI. Scale bars, 20 µm. Individual nuclei displaying representative categories of stains are shown to the right of each embryo. Embryos ranged in size from 39 to 100 nuclei. (E) Bar plot of percentage of nuclei with coincident accumulation of Xist RNA and H3K27me3 in individual *Eed*^fl/fl and *Eed*^-/- embryos. Each bar is an individual embryo. Embryo numbers under the bars correspond to the same embryos plotted in F). (F) Bar plots of percentage of nuclei with or without Xist RNA-coating and Rnf12 RNA expression in the embryos stained in D) and plotted in E). The numbers under the bars correspond to the same embryos plotted in E).
DOI: https://doi.org/10.7554/eLife.44258.008
The following figure supplement is available for figure 3:

**Figure supplement 1.** X-linked gene expression analysis in *Eed*^-/- embryos.
DOI: https://doi.org/10.7554/eLife.44258.009

averaged, however, the difference between *Eed*^m-/- and *Eed*^mz-/- embryos did not reach statistical significance (p=0.14) (**Figure 4C**; **Supplementary file 3**).

The shift in the ratio of X-linked gene expression towards the paternal allele in *Eed*^m-/- and *Eed*^mz-/- embryos could be due to increased paternal X-linked gene expression or to decreased maternal X-linked gene expression. To determine the source of the expression change, we calculated the normalized expression of genes on the maternal and paternal X-chromosomes for all genotypes (**Figure 4D** and **Figure 4—figure supplement 1C**). Whereas paternal X-linked genes significantly increased in expression, maternal X-linked gene expression decreased in *Eed*^m-/- and *Eed*^mz-/- embryos compared to *Eed*^fl/fl, *Eed*^fl/-, and *Eed*^-/- embryos. The increase in paternal X-linked gene expression in *Eed*^m-/- and *Eed*^mz-/- embryos was significant when compared to the three other genotypes. The decrease in maternal X-linked gene expression in *Eed*^m-/- and *Eed*^mz-/- embryos reached significance only vs. *Eed*^fl/fl embryos and not vs. *Eed*^fl/- and *Eed*^-/- embryos. The lack of a significant decrease between *Eed*^m-/- and *Eed*^mz-/- embryos compared to *Eed*^fl/- and *Eed*^-/- embryos is likely due to the greater variation in maternal X-linked gene expression in *Eed*^fl/- and *Eed*^-/- embryos (**Supplementary file 3**). Finally, *Eed*^mz-/- embryos displayed a significant increase in paternal X-linked gene expression compared to *Eed*^m-/- embryos (p=0.02; **Supplementary file 2** and **Supplementary file 3**), suggesting that zygotic EED can contribute to the silencing of a subset of X-linked genes in blastocysts.

To validate the *Eed*^m-/- and *Eed*^mz-/- blastocyst RNA-Seq data, we again analyzed allele-specific expression of *Xist*, *Rnf12*, *Atrx*, and *Pgk1* in E3.5 blastocysts by Pyrosequencing. Pyrosequencing also showed a significant defect in the initiation and establishment of imprinted X-inactivation in *Eed*^m-/- and *Eed*^mz-/- embryos (**Figure 4E** and **Figure 4—figure supplement 1D**; **Supplementary file 4**). In *Eed*^m-/- and *Eed*^mz-/- embryos, *Xist* expression unexpectedly increased from the maternal-X relative to the paternal-X. Conversely, the expression of *Rnf12* and *Atrx* increased from the paternal-X relative to the maternal-X in *Eed*^m-/- embryos. In *Eed*^mz-/- embryos, in addition to *Rnf12* and *Atrx*, *Pgk1* also displayed nearly equal levels of expression from the maternal and paternal alleles. The Pyrosequencing results thus recapitulate the defects in imprinted X-inactivation observed by RNA-Seq.

Together, the RNA-Seq and Pyrosequencing data lead to several suggestions. The first is that maternal EED depletion in the oocyte induces *Xist* from the maternal X-chromosome in the early embryo. This derepression is consistent with maternal PRC2 repressing the maternal *Xist* locus, which is marked by H3K27me3 in the oocyte [**Figure 4—figure supplement 1E**; (**Zheng et al., 2016**). Ectopic *Xist* induction from the maternal-X then results in the silencing of genes on that X-chromosome. The second major suggestion is that loss of maternal EED induces paternal X-linked genes. Finally, the data implicate zygotic EED expression in the silencing of a subset of paternal X-linked genes at the onset of imprinted X-inactivation.

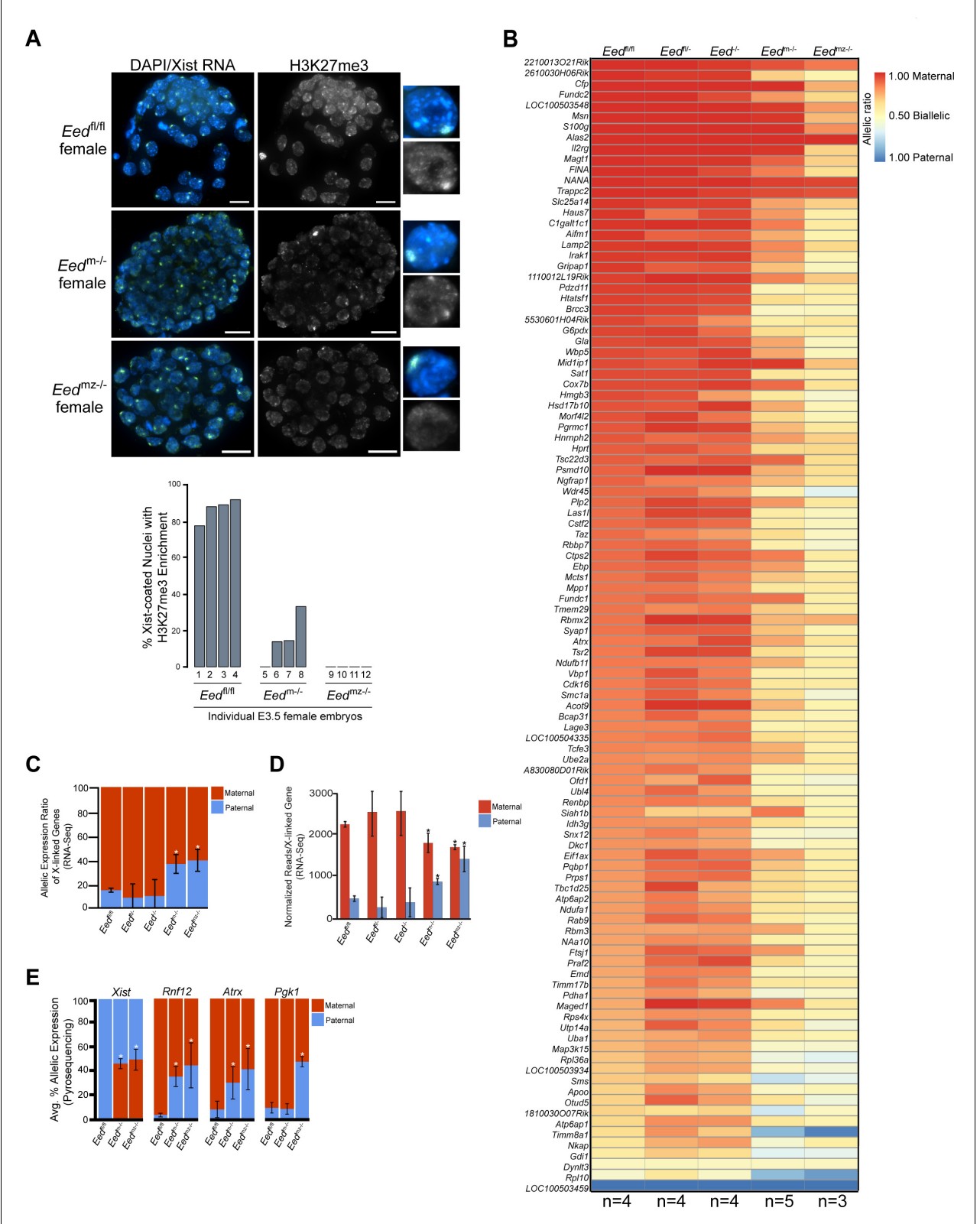

**Figure 4.** Defective imprinted X-inactivation initiation in blastocysts lacking maternal EED. See also *Figure 4—figure supplement 1*. (**A**) RNA FISH detection of Xist RNA (green) and IF stain for H3K27me3 (white) in representative *Eed*^m-/- and *Eed*^mz-/- female blastocysts. Nuclei are stained blue with DAPI. Scale bars, 20 μm. *Eed*^fl/fl blastocyst from *Figure 3D* shown for comparison. Right, individual representative nuclei. Mutant embryos ranged in size from 46 to 80 nuclei. Bar plot shows percentage of nuclei in each embryo analyzed that displayed H3K27me3 enrichment on the Xist RNA-coated

*Figure 4 continued on next page*

*Figure 4 continued*

X-chromosome. (B) Maternal:paternal X-linked gene expression heat map of female *Eed*^m-/- and *Eed*^mz-/- blastocysts. Five *Eed*^m-/- and three *Eed*^mz-/- embryos were sequenced individually and only genes with informative allelic expression in all samples are plotted (see Materials and methods). *Eed*^fl/fl, *Eed*^fl/-, and *Eed*^-/- data from **Figure 3A** shown for comparison. Genes are ordered on the basis of allelic expression in *Eed*^fl/fl embryos. (C) Average maternal:paternal X-linked gene expression ratio from the RNA-Seq data shown in B). *Eed*^fl/fl, *Eed*^fl/-, and *Eed*^-/- data from **Figure 3B** shown for comparison. The mean allelic expression of X-linked genes is significantly different between *Eed*^m-/- and *Eed*^fl/fl, and *Eed*^mz-/- and *Eed*^fl/fl blastocysts. (p<0.05, Welch's two-sample T-test). Pairwise statistical comparisons between all genotype groups are included in **Supplementary file 3**. (D) Average normalized maternal and paternal X-linked gene expression in blastocysts. Maternal and paternal X-linked gene expression is significantly different between *Eed*^m-/- and *Eed*^mz-/- embryos compared to *Eed*^fl/fl embryos (*, p<0.05, Two-tailed Student's T-test). Pairwise statistical comparisons between all genotypes are included in **Supplementary file 3**. (E) Pyrosequencing-based quantification of allelic expression of X-linked genes in *Eed*^m-/- and *Eed*^mz-/- blastocysts. *Eed*^fl/fl data from **Figure 3C** are shown for comparison. Error bars represent the standard deviation of data from 3 to 6 independent blastocyst embryos. The mean allelic expression of *Xist*, *Rnf12*, and *Atrx* is significantly different between *Eed*^fl/fl and *Eed*^m-/- embryos. The mean allelic expression of *Xist*, *Rnf12*, *Pgk1*, and *Atrx* is significantly different between *Eed*^fl/fl and *Eed*^mz-/- embryos (p<0.05, Welch's two-sample T-test). Pairwise statistical comparisons for all genes and between all genotypes are included in **Supplementary file 4**.

DOI: https://doi.org/10.7554/eLife.44258.010

The following figure supplement is available for figure 4:

**Figure supplement 1.** Generation and X-linked gene profiling of *Eed*^m-/- and *Eed*^mz-/- embryos.

DOI: https://doi.org/10.7554/eLife.44258.011

## Maternal EED silences *Xist* on the maternal-X

To validate the RNA-Seq and Pyrosequencing data from the maternal *Eed* mutants, we performed RNA FISH in *Eed*^m-/- and *Eed*^mz-/- blastocysts for *Xist* and *Rnf12* (**Figure 5A**). Whereas most nuclei in *Eed*^m-/- and *Eed*^mz-/- females displayed a single Xist RNA coat and monoallelic expression of *Rnf12*, a subset displayed Xist RNA coating of both X-chromosomes. The majority of these nuclei also lacked *Rnf12* expression, suggesting silencing of *Rnf12* on both X-chromosomes.

We similarly examined *Eed*^mz-/- male blastocysts (**Figure 5B**). A subset of nuclei in *Eed*^mz-/- male mutant embryos also exhibited ectopic Xist RNA coating of their sole, maternally-inherited X-chromosome. Interestingly, *Eed*^mz-/- male embryos were present in two distinct morphological classes. The first category was comprised of large, well-developed embryos, which displayed few or no nuclei with Xist RNA coating. The second category consisted of underdeveloped embryos, which displayed Xist RNA-coating in much higher proportions (20–60% of nuclei). In both sets of embryos, Xist RNA coating was often accompanied by a loss of *Rnf12* expression from the Xist RNA-coated X-chromosome. These data suggest that Xist RNA coating hinders developmental progression by silencing genes on the ectopically Xist RNA-coated X-chromosome. *Eed*^mz-/- embryos that adaptively repress *Xist* may overcome this developmental deficiency.

The correlation between reduced frequency of ectopic Xist RNA-coated nuclei and development of *Eed*^mz-/- embryos led us to test the developmental competency of maternal-null *Eed* embryos. We assessed if *Eed*^m-/- embryos could yield live born animals. To our surprise, a small number of *Eed*^m-/- female as well as male embryos could live to term (**Figure 5C**), suggesting that the ectopic Xist RNA expression and coating could be resolved in maternal-null embryos of both sexes. Interestingly, significantly more females were born compared to males (p=0.02, Two-tailed Student's T-test), suggesting that females can more robustly extinguish ectopic Xist RNA expression compared to males. These data further suggest that zygotic EED expression is sufficient to compensate for the absence of maternal EED in a subset of the early embryos. *Eed*^mz-/- embryos are expected to be inviable, since loss of zygotic *Eed* expression results in lethality of both female and male embryos (**Faust et al., 1995**; **Shumacher et al., 1996**; **Wang et al., 2001**).

## Switching of imprinted to random X-inactivation in *Eed*^m-/- embryos

The relative paucity of ectopic Xist RNA-coated nuclei in female *Eed*^m-/- and *Eed*^mz-/- blastocysts observed by RNA FISH in **Figure 5A–B** is inconsistent with the robust ectopic Xist RNA expression from and silencing of maternal X-linked genes and the increased expression of paternal X-linked genes that were detected via Pyrosequencing and RNA-Seq (**Figure 4B–D**). We thus postulated that instead of undergoing imprinted inactivation of the paternal X-chromosome, *Eed*^m-/- and *Eed*^mz-/- blastocysts switch to random X-inactivation of either the maternal- or the paternal-X in individual cells. Such mosaicism would explain the silencing of maternal X-linked genes and the induction of

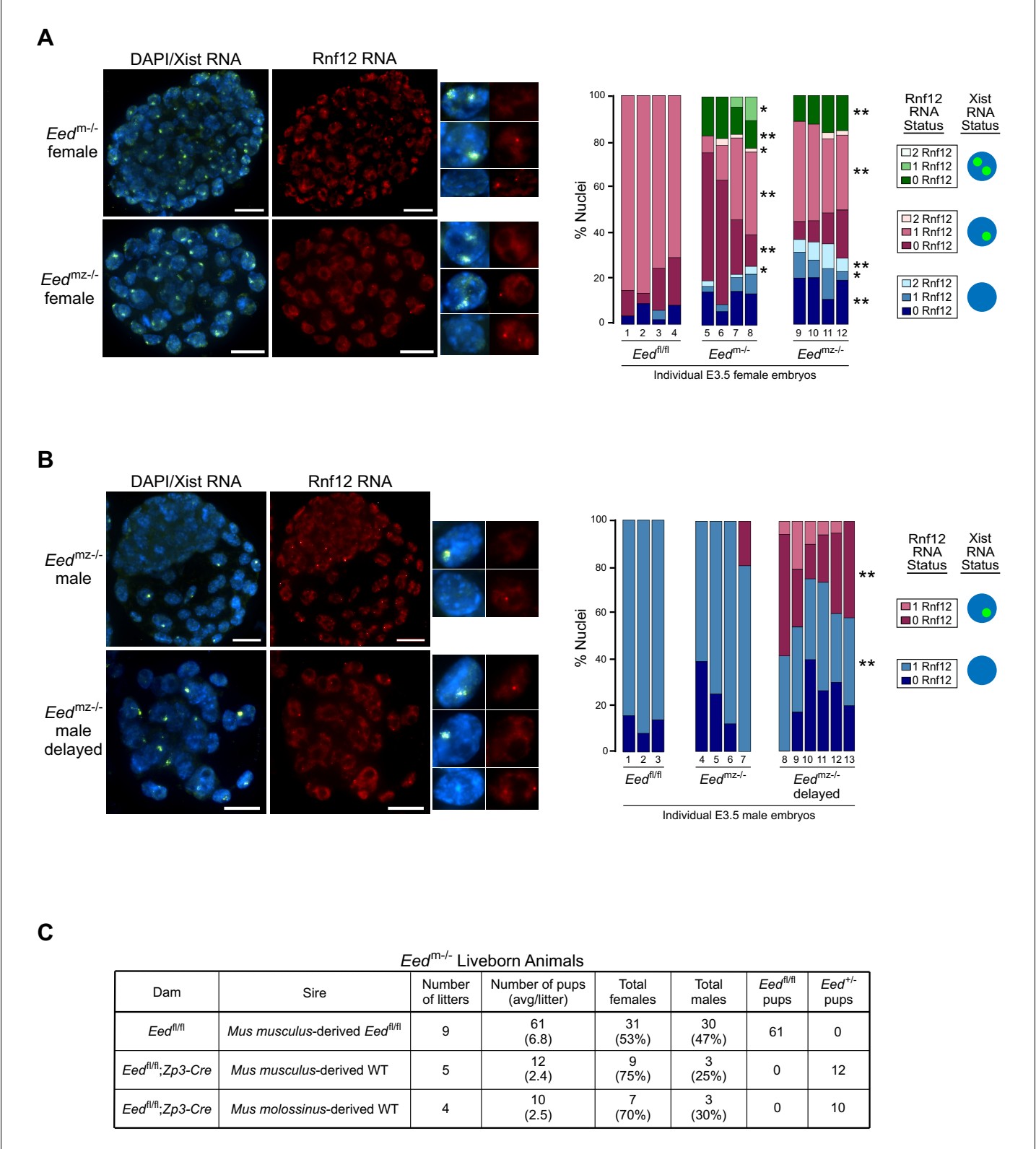

**Figure 5.** RNA FISH analysis of X-inactivation in *Eed*[m-/-] and *Eed*[mz-/-] blastocysts. (**A,B**) RNA FISH detection of Xist RNA (green) and Rnf12 RNA (red) in representative *Eed*[m-/-] and *Eed*[mz-/-] female (**A**) and *Eed*[mz-/-] male (**B**) blastocysts. Nuclei are stained blue with DAPI. Scale bars, 20 μm. Individual nuclei of representative categories of stain are shown to the right of each embryo. *Eed*[fl/fl] female data from **Figure 3D** shown for comparison. Mutant female embryos ranged in size from 46 to 80 nuclei. Fully developed mutant male embryos ranged in size from 53 to 110 nuclei. Delayed mutant male embryos

*Figure 5 continued on next page*

*Figure 5 continued*

ranged in size from 30 to 40 nuclei. Bar plot shows percentage of nuclei in each embryo with Xist RNA coats and/or Rnf12 RNA expression. Each bar represents an individual embryo and embryo numbers under the bars correspond to the same female embryos plotted in *Figure 4A*. *, p<0.05; **, p<0.01, Two-tailed Student's T-test, between $Eed^{m-/-}$ and $Eed^{fl/fl}$, or $Eed^{mz-/-}$ and $Eed^{fl/fl}$. (C) Data showing the number of $Eed^{m-/-}$ embryos which can live to term compared to $Eed^{fl/fl}$ embryos. WT, wild-type. Table shows $Eed^{m-/-}$ litters sired by *Mus musculus*-derived male or *Mus molossinus*-derived male. Male $Eed^{m-/-}$ offspring are underrepresented compared to females, p=0.02, Two-tailed Student's T-test.

DOI: https://doi.org/10.7554/eLife.44258.012

paternal X-linked gene expression in $Eed^{m-/-}$ and $Eed^{mz-/-}$ female embryos detected by RNA-Seq and Pyrosequencing.

To test the above model of X-inactivation mosaicism, we developed and applied an allele-specific Xist RNA FISH strategy on hybrid control $Eed^{fl/+}$ and test $Eed^{m-/-}$ female E3.5 blastocysts (Materials and methods; *Figure 6—figure supplement 1*). Allele-specific Xist RNA FISH allowed us to discriminate Xist RNA expression from the maternal vs. the paternal X-chromosome in individual cells. Allele-specific Xist RNA FISH displayed Xist RNA expression from only the paternal-X in $Eed^{fl/+}$ female blastocysts (*Figure 6A*), as would be expected from embryos stably undergoing imprinted X-inactivation of the paternal-X. In $Eed^{m-/-}$ female blastocysts, however, we saw a mosaic distribution of Xist RNA expression and coating. Whereas some $Eed^{m-/-}$ blastocyst nuclei displayed Xist RNA expression from and coating of the maternal-X, others exhibited Xist RNA expression from and coating of the paternal-X. A subset of nuclei in $Eed^{m-/-}$ blastocysts exhibited Xist RNA expression from both the maternal and paternal X-chromosomes (*Figure 6A*), consistent with the non-allele specific Xist RNA FISH data from $Eed^{m-/-}$ and $Eed^{mz-/-}$ blastocysts in *Figure 5A*. Male $Eed^{m-/-}$ embryos similarly displayed ectopic Xist RNA expression from and coating of their sole maternally-inherited X-chromosome in approximately 50% of nuclei (*Figure 6B*).

From the blastocyst data, we extrapolated that earlier $Eed^{m-/-}$ embryos may harbor a higher proportion of cells with ectopic Xist RNA coating of the maternal-X. This pattern would be later resolved into the mosaic Xist RNA coating pattern observed at the blastocyst stage in females and loss of the Xist RNA coat in males. We therefore performed allele-specific Xist RNA FISH on 3–16 cell control $Eed^{fl/+}$ and test $Eed^{m-/-}$ hybrid embryos. In the $Eed^{fl/+}$ female embryos, Xist RNA was expressed from and coated only the paternal X-chromosome (*Figure 7A*). Most $Eed^{m-/-}$ female embryos, by contrast, displayed a high percentage of nuclei with Xist RNA expression and coating of both X-chromosomes (*Figure 7A*). In male 3–17 cell embryos, $Eed^{fl/+}$ embryos did not show any nuclei with Xist RNA coating (*Figure 7B*). In $Eed^{m-/-}$ male embryos, by contrast, almost every nucleus exhibited ectopic *Xist* expression from and coating of the maternally-inherited X-chromosome (*Figure 7B*). Thus, in the absence of maternal EED most cells express *Xist* from both X-chromosomes in early female embryos and from the sole X in early male embryos. By the blastocyst stage, however, one of the two *Xist* alleles is stochastically silenced in most female cells and the sole *Xist* allele is silenced in most male cells.

## Lack of maternal EED in human embryos

Intriguingly, the Xist RNA coating of both X-chromosomes in female and of the single X in male early preimplantation $Eed^{m-/-}$ and $Eed^{mz-/-}$ mouse embryos resemble the pattern observed in preimplantation human female and male embryos (*Okamoto et al., 2011*; *Petropoulos et al., 2016*). In early preimplantation human embryos, females display Xist RNA coating of both Xs and males of their sole maternally-inherited X-chromosome. We therefore hypothesized that the Xist RNA expression profile in early human embryos may reflect the absence of maternally-derived EED and other core PRC2 proteins in human oocytes. To test this hypothesis, we analyzed RNA-Seq data from mouse and human oocytes to determine the expression levels of core PRC2 genes *Eed*, *Ezh2*, *Ezh1*, and *Suz12* (*Kobayashi et al., 2012*; *Macfarlan et al., 2012*; *Reich et al., 2011*). Compared to mouse oocytes, human oocytes expressed all four genes at negligible levels (*Figure 8A*). This difference in the expression of PRC2 components in oocytes may underlie why early mouse but not human embryos undergo imprinted X-inactivation.

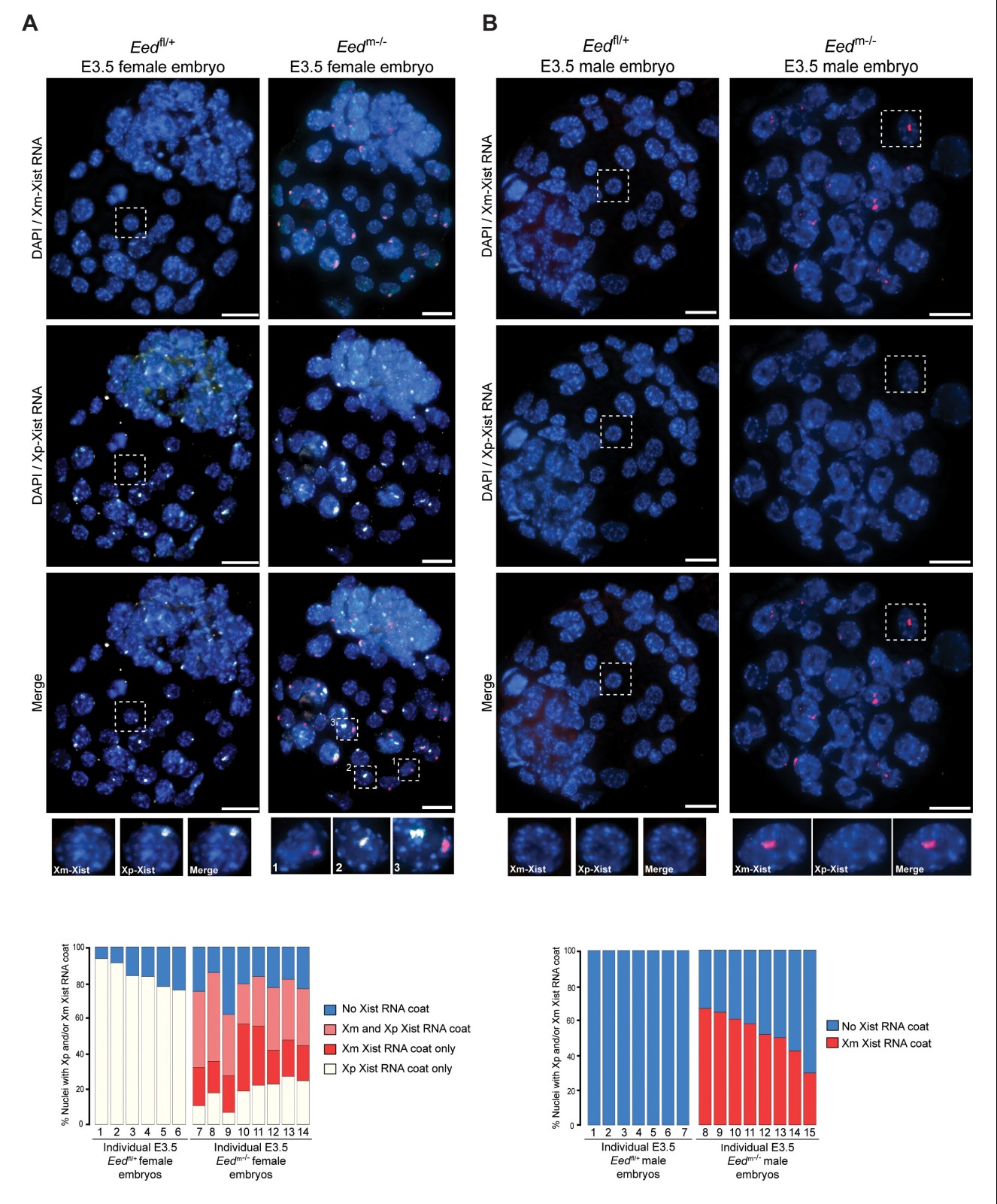

**Figure 6.** Switching of imprinted to random X-inactivation in E3.5 embryos lacking maternal EED. See also *Figure 6—figure supplement 1*. (A,B) Allele-Specific Xist RNA FISH in *Eed*^fl/+ and *Eed*^m-/- male and female E3.0-E3.5 blastocyst embryos. Xist RNA expressed from the maternal X-chromosome is indicated in red and from the paternal X-chromosome in white. Representative embryos are depicted. Nuclei are stained blue with DAPI. Scale bars, 20 μm.

*Figure 6 continued on next page*

*Figure 6 continued*

DOI: https://doi.org/10.7554/eLife.44258.013
The following figure supplement is available for figure 6:

**Figure supplement 1.** Characterization of allele-specific Xist RNA FISH probe in cells and embryos.
DOI: https://doi.org/10.7554/eLife.44258.014

## Discussion

Genomic imprinting is a paradigm of transgenerational epigenetic inheritance, since the two parental alleles undergo diametrically divergent transcriptional fates in a parent-of-origin-specific manner in the embryo. Imprinted X-inactivation is an extreme example of genomic imprinting in that most genes on the paternally-inherited X-chromosome undergo silencing. The maternal X-chromosome, by contrast, remains active. Here, we test the role of core PRC2 protein EED in the initiation of imprinted X-inactivation during early mouse embryogenesis. We defined the transition of maternal to zygotic EED expression in the early embryo and found the presence of maternal and a relative absence of zygotic EED when imprinted X-inactivation begins. Upon ablation of *Eed* in the oocyte and the absence of maternally-derived EED in the embryo, the initiation of imprinted X-inactivation is compromised (*Figure 8B*). Maternal-null (*Eed*$^{m-/-}$ and *Eed*$^{mz-/-}$) but not zygotic-null (*Eed*$^{-/-}$) early preimplantation female and male embryos ectopically induced Xist RNA from the maternal X-chromosome. Early *Eed*$^{m-/-}$ female embryos therefore display Xist RNA-coating of both X-chromosomes and *Eed*$^{m-/-}$ mutant males of the sole maternally-inherited X-chromosome.

PRC2-catalyzed H3K27me3 marks the *Xist* locus on the maternal X-chromosome during oogenesis (*Zheng et al., 2016*). In agreement, the injection of the H3K27me3 demethylase *Kdm6b* in the zygote resulted in the derepression of the *Xist* locus on the maternal X-chromosome in 8–16 cell embryos (*Inoue et al., 2017*). Female morulas derived from *Kdm6b*-injected zygotes displayed Xist RNA coating of both the maternal and the paternal X-chromosome in most blastomeres, suggesting inactivation of both Xs in the embryo. Nullizygosity of X-linked gene expression due to inactivation of both Xs in females or of the single-X in males is expected to result in cell and embryo lethality (*Gayen et al., 2015*). The conditional deletion of *Eed* in the oocyte, however, yielded live born mice, implying that ectopic *Xist* expression due to H3K27me3 loss and the ensuing inactivation of the maternal-X in the early embryo is resolved later [this study; (*Prokopuk et al., 2018*)]. In agreement, our study shows that by the blastocyst stage most nuclei in *Eed*$^{m-/-}$ and *Eed*$^{mz-/-}$ female embryos exhibit only one Xist RNA coat. However, instead of Xist RNA coating exclusively of the paternal X-chromosome as in WT embryos, the maternal *Eed* mutants express Xist RNA from and coat either the maternal or the paternal X-chromosome, a hallmark of random X-inactivation. This randomization persists later in development in extraembryonic tissues (data not shown), which normally maintain imprinted inactivation of the paternal-X. Like *Eed*$^{m-/-}$ and *Eed*$^{mz-/-}$ females, *Eed*$^{m-/-}$ and *Eed*$^{mz-/-}$ male blastocysts also extinguish ectopic *Xist* induction.

In addition to maternal EED, our data argue that zygotically generated EED contributes to imprinted X-inactivation in the early embryo. In comparison to *Eed*$^{m-/-}$ embryos, *Eed*$^{mz-/-}$ female blastocysts displayed a further increase in paternal X-linked gene expression. One interpretation of these data is that the onset of zygotic EED expression results in the preferential installation of H3K27me3 at the *Xist* locus on the maternal-X in some cells of early *Eed*$^{m-/-}$ embryos. These cells thus forestall or extinguish *Xist* expression from the maternal X-chromosome and inactivate the paternal-X, ultimately resulting in more cells in the embryo in which the paternal-X is inactive compared to the maternal-X. Loss of both maternal and zygotic EED would annul such biased inactivation of the paternal-X and thereby cause a greater increase in paternal X-linked gene expression in *Eed*$^{mz-/-}$ vs. *Eed*$^{m-/-}$ embryos. An alternative possibility is that zygotic EED functions to maintain silencing preferentially of paternal X-linked genes in the early embryo. The differential sensitivity of genes on the maternal vs. paternal X-chromosomes to zygotic EED in *Eed*$^{m-/-}$ embryos may reflect the different kinetics of inactivation of the two X-chromosomes. The ectopic induction of *Xist* and X-linked gene silencing on the maternal-X may occur more slowly compared to that on the paternal-X. Due to this delay, genes on the maternal-X would still be in the process of undergoing silencing in *Eed*$^{m-/-}$ blastocysts. A subset of paternal X-linked genes, on the other hand, may have established silencing and are now in the maintenance phase of X-inactivation in the blastocysts. In the absence

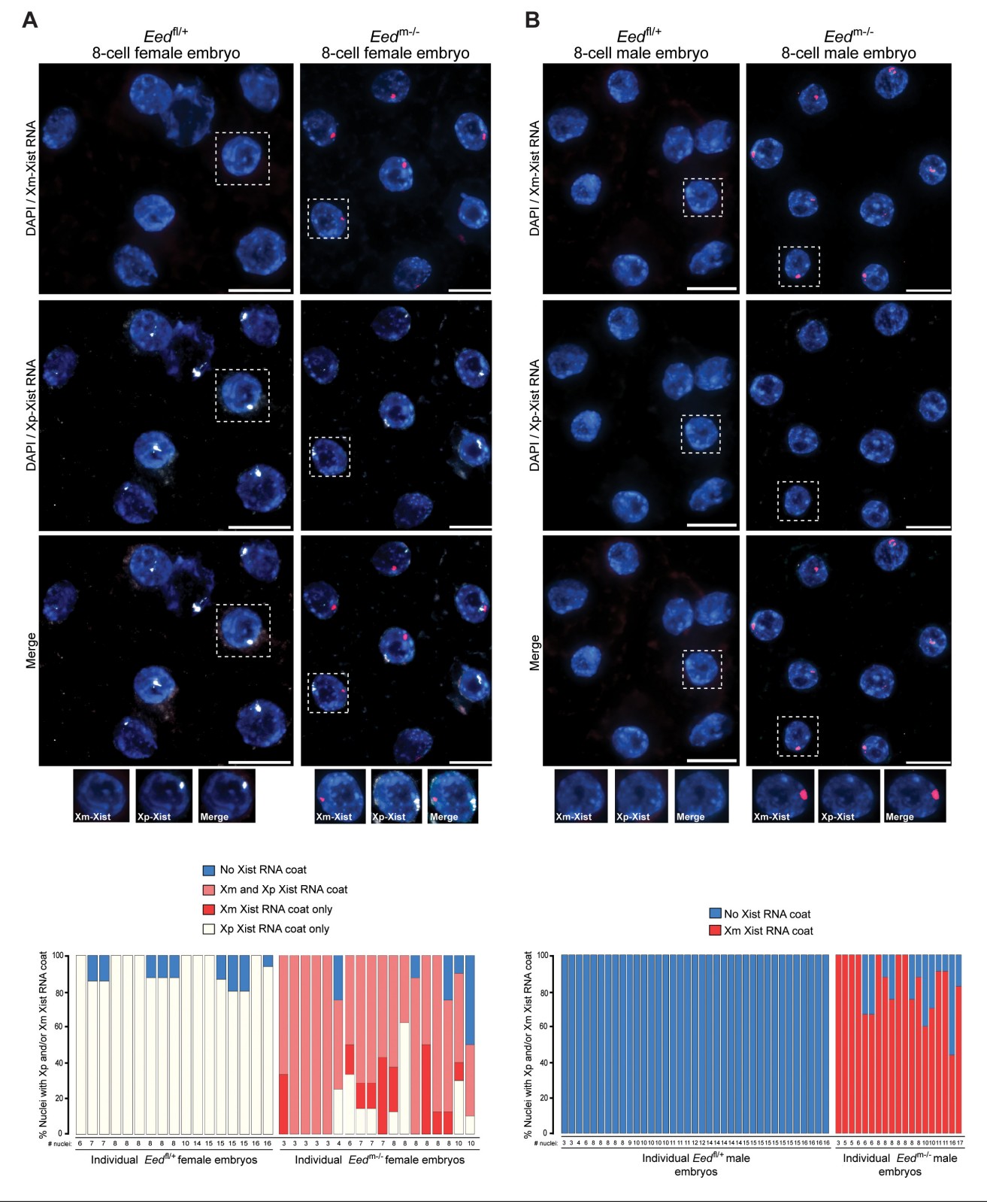

**Figure 7.** Switching of imprinted to random X-inactivation in 3–16 cell embryos lacking maternal EED. (A,B) Allele-Specific Xist RNA FISH in $Eed^{fl/+}$ and $Eed^{m-/-}$ female and male 3–16 cell embryos. Xist RNA expressed from the maternal X-chromosome is indicated in red and from the paternal X-chromosome in white. Representative embryos are depicted. Nuclei are stained blue with DAPI. Scale bars, 20 μm.

DOI: https://doi.org/10.7554/eLife.44258.015

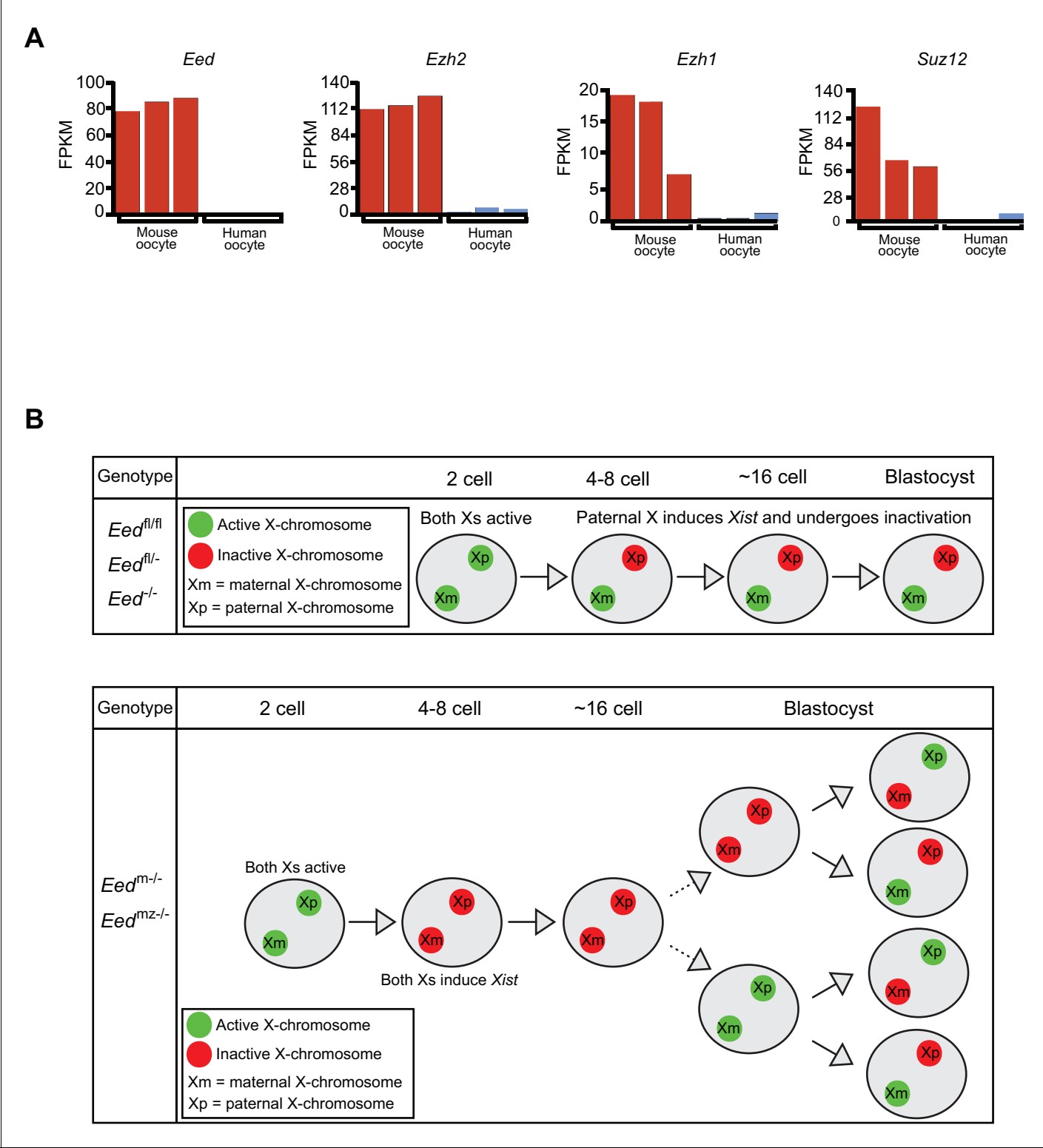

**Figure 8.** Lack of PRC2 expression in human oocytes and a path to randomization of X-inactivation in early embryos. (**A**) Expression levels by RNA-Seq of core PRC2 components in human and mouse oocytes. (**B**) Model of maternal PRC2 function during preimplantation mouse embryogenesis.
DOI: https://doi.org/10.7554/eLife.44258.016

of both maternal and zygotic EED, then, $Eed^{mz-/-}$ blastocysts fail to maintain silencing of these paternal X-linked genes. Our previous work has shown that zygotic EED is in fact required to maintain silencing of a discrete set of paternal X-linked genes during imprinted X-inactivation (*Kalantry and Magnuson, 2006*; *Kalantry et al., 2006*; *Maclary et al., 2017*).

The ability of the cells of early $Eed^{m-/-}$ and $Eed^{mz-/-}$ embryos to resolve Xist RNA coating of both Xs in females or of the single X in males implies that the early mouse embryo has an X-chromosome counting mechanism that ensures that a single X-chromosome remain active in females as well as in males, irrespective of its parent of origin. Such a counting mechanism has previously been proposed by Takagi and colleagues to explain the kinetics of Xist RNA induction in *XX* and *XY* androgenetic embryos, which harbor only paternal X-chromosomes (*Okamoto et al., 2000*). Like in $Eed^{m-/-}$ embryos, androgenetic 4 and 8–16 cell embryos also initially induce Xist RNA from all Xs, which is resolved at the blastocyst stage and results in females displaying a single Xist RNA coat in most nuclei and males exhibiting few or no nuclei with Xist RNA coating (*Okamoto et al., 2000*). Molecular sensing of the X-chromosomal complement in imprinted X-inactivation is also suggested by studies of diploid *XX* parthenogenetic or gynogenetic embryos, which harbor two maternal X-chromosomes. In these preimplantation bi-maternal *XX* embryos, *Xist* expression is delayed and appears to occur stochastically from one or the other X-chromosome (*Kay et al., 1994*). In agreement, the extraembryonic tissues of post-implantation *XX* parthenogenotes display hallmarks of random X-inactivation instead of the imprinted form observed in WT extraembryonic cells (*Rastan et al., 1980*). Randomization of X-inactivation in extraembryonic cells of mouse embryos with two paternal or maternal X-chromosomes led Takagi and colleagues to suggest that imprinted X-inactivation in placental mammals may have arisen from random X-inactivation (*Matsui et al., 2001*), a notion that our data from $Eed^{m-/-}$ and $Eed^{mz-/-}$ embryos agree with.

Evidence suggests that the X-linked *Rnf12* gene may be a key component of the X-chromosome counting mechanism during imprinted X-inactivation. The maternal-X allele of *Rnf12* is required to induce *Xist* from the paternal-X in preimplantation mouse embryos (*Shin et al., 2010*). Upon Xist RNA coating, *Rnf12* is rapidly silenced on the paternal X-chromosome (*Kalantry et al., 2009*; *Namekawa et al., 2010*; *Patrat et al., 2009*). In $Eed^{m-/-}$ and $Eed^{mz-/-}$ embryos, in addition to the paternal *Rnf12* allele, the maternal *Rnf12* allele is also stringently silenced due to ectopic Xist RNA coating of the maternal-X. Since *Rnf12* is required for Xist RNA induction in the preimplantation embryo, the silencing of all *Rnf12* alleles in $Eed^{m-/-}$ and $Eed^{mz-/-}$ female and male embryos may paradoxically lead to the loss of Xist RNA expression from both Xs in females or from the sole X-chromosome in males. In females, this transient state of two active-Xs may then be followed by random X-inactivation, analogously to how differentiating pluripotent epiblast cells undergo random X-inactivation (*Gayen et al., 2015*; *Maclary et al., 2014*; *Mak et al., 2004*). The X-chromosome counting process and randomization of X-inactivation in the early embryo may explain how $Eed^{m-/-}$ embryos can yield live born animals [this study and (*Prokopuk et al., 2018*).

In the course of preparing this manuscript, a publication reported that extraembryonic tissues of *Eed* maternal-null female post-implantation embryos exhibit random X-inactivation (*Inoue et al., 2018*). The primary piece of data in the study supporting this conclusion is the expression of maternal and paternal X-linked genes, including *Xist*, in post-implantation E6.5 female $Eed^{m-/-}$ extraembryonic tissues by allele-specific RNA-Seq. Although in agreement with our conclusions, the study did not directly demonstrate when imprinted X-inactivation switches to random X-inactivation and whether loss of zygotic *Eed* would result in a similar outcome. Our study, by contrast, genetically dissects the relative contributions of maternal vs. zygotic EED in the initiation and establishment of imprinted X-inactivation. We are thus able to pinpoint when and how the loss of maternal EED converts imprinted X-inactivation to random X-inactivation in preimplantation embryos. Genetically testing the requirement of maternal vs. zygotic EED is necessary to determine that the establishment of imprinted X-inactivation in the preimplantation embryo is maternally but not zygotically controlled.

Xist RNA expression in $Eed^{m-/-}$ mouse embryos mimics the pattern observed in human embryos, which do not undergo imprinted X-inactivation and ultimately display only random X-inactivation (*Okamoto et al., 2011*; *Petropoulos et al., 2016*). In agreement, like the $Eed^{m-/-}$ and $Eed^{mz-/-}$ oocytes, human oocytes are devoid of expression of *Eed*, as well as expression of the other core PRC2 genes, suggesting that the presence or absence of maternal PRC2 or related chromatin modifying proteins may dictate whether placental mammals undergo imprinted X-inactivation.

# Materials and methods

**Key resources table**

| Reagent type (species) or resource | Designation | Source or reference | Identifiers | Additional information |
|---|---|---|---|---|
| Gene (*Mus*) | *Eed* | ENSEMBL | ENSEMBL: ENSMUSG00000030619 | Chromosome 7: 89,954,654–89,980,983 reverse strand |
| Strain, strain background (*Mus molossinus*) | JF1/Ms; *Mus molossinus* | JAX | JAX:003720; RRID:MGI:2164136 | |
| Strain, strain background (*Mus musculus*) | 129/S1; *Mus musculus* | JAX | JAX:002448 | |
| Genetic reagent (*Mus musculus*) | *Eed*^fl | *Maclary et al. (2017)* | | Lox sites inserted into the *Eed* gene in introns surrounding exon 7. |
| Genetic reagent (*Protamine-cre*) | *Prm-cre* | *O'Gorman et al. (1997)*; JAX | JAX:003328 | |
| Genetic reagent (*Zp3-cre*) | *Zp3-cre* | *Lewandoski et al. (1997)*; JAX | JAX:003651 | |
| Biological sample (mouse embryo) | *Mus musculus*; *Mus molossinus* | this paper | | 2 cell stage to blastocyst stage embryos |
| Biological sample (RNA) | *Mus musculus*; *Mus molossinus* | this paper | | Generated from female blastocysts |
| Antibody | Monoclonal EED (Rabbit monoclonal) | *Sewalt et al. (1998)* | | Obtained from Otte Lab; Dilution: *Figure 1* –1:1000, *Figure 2* –1:2500 |
| Antibody | Polyclonal H3K27me3 (Rabbit polyclonal) | Millipore | Millipore:#ABE44 | Dilution: *Figure 1* – 1:5000, *Figure 2* – 1:25000 |
| Antibody | Alexa Fluor DαM 555 (secondaries) | Invitrogen | Invitrogen:#A32773 | Dilution: *Figure 1* –1:300, *Figure 2* – 1:500 |
| Antibody | Alexa Fluor DαRb 488 (secondaries) | Invitrogen | Invitrogen:#A21206 | Dilution: *Figure 1* –1:300, *Figure 2* – 1:500 |
| Antibody | Alexa Fluor DαRb 647 (secondaries) | Invitrogen | Invitrogen:#A31573 | Dilution: 1:300 |
| Sequence-based reagent | Quasar dye 570 | Biosearch Technologies | primer sequences in *Supplementary file 5* | Allele-specific probe dye; labeled *M. musculus*-specific oligo |
| Sequence-based reagent | Quasar dye g70 | Biosearch Technologies | primer sequences in *Supplementary file 5* | Allele-specific probe dye; labeled *M. molossinus*-specific oligo |
| Commercial assay or kit | Dynabeads mRNA DIRECT Kit | Thermo Fisher | ThermoFisher:#610.11 | |
| Commercial assay or kit | Takara SMARTer Seq V4 stranded low input kit | Takara | Takara:#634889 | |

*Continued on next page*

*Continued*

| Reagent type (species) or resource | Designation | Source or reference | Identifiers | Additional information |
|---|---|---|---|---|
| Commercial assay or kit | BioPrime DNA Labeling System | Invitrogen | Invitrogen: #18094011 | |
| Software, algorithm | FastQC | http://www.bioinformatics.babraham.ac.uk/projects/fastqc | RRID:SCR_014583 | |
| Software, algorithm | R | https://www.r-project.org | RRID:SCR_001905 | Used in RNA-Seq analysis |
| Software, algorithm | VCFtools | *Danecek et al. (2011)* | RRID:SCR_001235 | Used in RNA-Seq analysis |
| Software, algorithm | STAR | *Dobin et al. (2013)* | RRID:SCR_015899 | Used in RNA-Seq analysis |
| Software, algorithm | HTSeq | *Anders et al. (2015)* | RRID:SCR_005514 | Used in RNA-Seq analysis |
| Software, algorithm | FeatureCounts | *Liao et al. (2014)* | RRID:SCR_012919 | Used in RNA-Seq analysis |
| Other | DAPI stain | Invitrogen | Invitrogen: #D21490 | Dilution: 1:250,000 |
| Other | Cy3-dCTP | GE Healthcare | GEHealthcare: #PA53021 | |
| Other | Fluorescein-12-UTP | Roche | Roche: #11427857910 | |
| Other | Cy5-CTP | GE Healthcare | GEHealthcare: #25801087 | |
| Other | SSC | Ambion | Ambion: #AM9765 | RNA FISH hybridization buffer, working concentation: 4X; allele-specific RNA FISH, working concentration: 2X |
| Other | Dextrane sulfate | Millipore | Millipore:#S4030 | RNA FISH hybridization buffer, working concentation: 20%; allele-specific RNA FISH, working concentration: 10% |
| Other | Formamide, deionized | VWR Life Sciences | VWR:#0606 | RNA FISH hybridization buffer; AlSp working concentration: 10% |
| Other | BSA | New England Biolabs | NEB:#B9001S | IF blocking buffer, working concentration: 0.5 mg/ml |
| Other | yeast tRNA | Invitrogen | Invitrogen: #15401–029 | IF blocking buffer, working concentration: 50 ug/ml |
| Other | RNAase out | Invitrogen | Invitrogen: #10777–019 | IF blocking buffer, working concentration: 80 units/ml |
| Other | Tween-20 | Thermo Fisher | ThermoFisher:# BP337-100 | IF blocking buffer, working concentration: 0.2% |

*Continued on next page*

*Continued*

| Reagent type (species) or resource | Designation | Source or reference | Identifiers | Additional information |
|---|---|---|---|---|
| Other | PBS | Gibco | Gibco:#14200 | IF blocking buffer, working concentation: 1X |
| Other | Vectashield | Vector Labs | VectorLabs: #H-1000 | Mounting medium for IF/ RNA FISH samples |

## Ethics statement

This study was performed in strict accordance with the recommendations in the Guide for the Care and Use of Laboratory Animals of the National Institutes of Health. All animals were handled according to protocols approved by the University Committee on Use and Care of Animals (UCUCA) at the University of Michigan (protocol #s PRO6455 and PRO8425).

## Mice

Mice harboring a conditional mutation in *Eed* were described in our prior publication (*Maclary et al., 2017*). A *Mus molossinus* JF1 X-chromosome was introgressed to generate $Eed^{fl/fl}$; $X^{JF1}Y$ males. *Mus musculus* $Eed^{fl/fl}$ females were backcrossed onto the 129/S1 background. The X-linked *Gfp* transgenic (*X-Gfp*) and JF1 strains have been described previously (*Hadjantonakis et al., 1998*; *Kalantry and Magnuson, 2006*; *Kalantry et al., 2006*; *Kalantry et al., 2009*; *Maclary et al., 2017*).

Embryos generated for the purpose of allele-specific RNA sequencing (RNA-Seq), Pyrosequencing, or allele-specific RNA fluorescence *in situ* hybridization (FISH) were sired by males harboring the $X^{JF1}$ X-chromosome. Embryos generated for immunofluorescence (IF) and non-allele specific RNA FISH were sired by males harboring the *X-Gfp* transgene. The paternal *X-Gfp* is only transmitted to daughters. Thus, GFP fluorescence conferred by the paternally-transmitted *X-Gfp* transgene was used to sex the embryos.

For derivation of embryos lacking zygotic *Eed*, the *Protamine-Cre* (*Prm-Cre*) transgene was bred into an $Eed^{fl/fl}$ or $Eed^{fl/-}$ background. *Prm-Cre* is expressed only during spermatogenesis (*O'Gorman et al., 1997*), thus resulting in the deletion of the *Eed* floxed allele in the male germline. For derivation of embryos lacking maternal EED, a *Cre* transgene controlled by the *Zona pellucida three* gene promoter (*Zp3-Cre*) (*Lewandoski et al., 1997*), was used to delete the floxed *Eed* alleles in growing oocytes.

## Mouse embryo dissections and processing

Embryonic day (E) 3.5 embryos were isolated essentially as described (*Maclary et al., 2014*). Embryos were flushed from the uterine limbs in 1X PBS (Invitrogen, #14200) containing 6 mg/ml BSA (Invitrogen, #15260037).

Two to sixteen cell embryos were flushed from oviducts of superovulated females with 1X PBS (Invitrogen, #14200) containing 6 mg/ml BSA (Invitrogen, #15260037) or M2 medium (Sigma, #M7167). For superovulation, 4–5 week-old, or 9–12 week-old females were treated with 5 IU of pregnant mare's serum gonadotropin (PMSG, Sigma, # G-4877) and 46 hr later with 5 IU of human chorionic gonadotropin (hCG, Sigma, #CG-5). Embryos were harvested 48–74 hr post hCG.

The zona pellucida surrounding embryos was removed through incubation in cold acidic Tyrode's solution (Sigma, #T1788), followed by neutralization through several transfers of cold M2 medium (Sigma, #M7167).

Isolated E3.5 embryos were either lysed for RNA isolation or plated onto 0.2% gelatin- (Sigma, #G2500) and/or 0.01% Poly-L-Lysine (PLL, Sigma, # P4707)-coated glass coverslips (22mm X 22mm, Thermo Fisher Scientific, #12548B) in 0.25X PBS for immunofluorescence (IF) coupled with RNA in situ hybridization (FISH). 2–16 cell embryos were plated on coverslips coated in 0.01% Poly-L-Lysine for IF. E3.5 or 4–16 cell embryos were plated on coverslips coated with 1X Denhardt's (Sigma, #D9905) solution for allele-specific RNA FISH. For plated embryos, excess solution was aspirated, and coverslips were air-dried for approximately 15–30 min. After drying, embryos were

permeabilized and fixed in 50 µL solution of either 0.05% or 0.1% Tergitol (Sigma, #NP407) with 1% paraformaldehyde (Electron Microscopy Sciences, #15710) in 1X PBS for 5 min, followed by 1% para-formaldehyde in 1X PBS for an additional 5 min. Excess solution was gently tapped off onto paper towels, and coverslips were rinsed 3X with 70% ethanol and stored in 70% ethanol at −20˚C prior to IF and/or RNA FISH.

## PCR

For embryo DNA isolation, embryos were isolated as described above. Individual blastocysts were lysed in 15 µL buffer composed of 50 mM KCl, 10 mM Tris-Cl (pH 8.3), 2.5 mM $MgCl_2$, 0.1 mg/mL gelatin, 0.45% NP-40, 0.45% Tween-20, and 0.4 mg/mL Proteinase K (Fisher, #BP1700). Embryos in lysis buffer were incubated at 50˚C overnight, then stored at 4˚C until use. Genomic PCR used 1–3 µL lysate per sample. Reactions for *Eed* were carried out in ChromaTaq buffer (Denville Scientific) with 2.5 mM $MgCl_2$ added. *XX* vs. *XY* sexing PCR reactions were carried out in Klentherm buffer (670 mM Tris pH 9.1, 160 mM $(NH_4)SO_4$, 35 mM $MgCl_2$,15mg/ml BSA). Both used RadiantTaq DNA polymerase (Alkali Scientific, #C109). Primer sequences are described in *Supplementary file 5*.

Liveborn animals from the cross of *Eed*^fl/fl^;*Zp3-Cre* female by WT male were genotyped for *Eed* to confirm deletion of the floxed allele. Ear punches were taken after weaning and lysed in 50 µL of lysis buffer (above). Ear punches were incubated at 50˚C overnight, then stored at 4˚C until use. 1 µL of DNA lysate was used per reaction. *Eed* PCRs were carried out as above.

## Quantification of allele-specific RNA expression by Pyrosequencing

Allele-specific expression was quantified using the Qiagen PyroMark sequencing platform, as previously described (*Gayen et al., 2015*). Briefly, the amplicons containing SNPs were designed using the PyroMark Assay Design software. cDNAs were synthesized using Invitrogen SuperScript III One-Step RT-PCR System (Invitrogen, #12574–026). Following the PCR reaction, 5 µL of the 25 µL reaction was run on a 3% agarose gel to assess the efficacy of amplification. The samples were then prepared for Pyrosequencing according to the standard recommendations for use with the PyroMark Q96 ID sequencer. All amplicons spanned intron(s), thus permitting discrimination of RNA vs. any contaminating genomic DNA amplification due to size differences. Control reactions lacking reverse transcriptase for each sample were also performed to rule out genomic DNA contamination. E3.5 embryos of similar sizes for all genotypes were used in the Pyrosequencing assays. Pyrosequencing primer sequences are described in *Supplementary file 5*.

## Immunofluorescence (IF)

Embryos mounted on gelatin-, PLL-, and/or PLL/gelatin-coated glass coverslips were washed 3 times in 1X PBS for 3 min each while shaking. Coverslips were then incubated in blocking buffer consisting of 0.5 mg/mL BSA (New England Biolabs, #B9001S), 50 µg/mL yeast tRNA (Invitrogen, #15401–029), 80 units/mL RNAseOUT (Invitrogen, #10777–019), and 0.2% Tween 20 (Fisher, #BP337-100) in 1X PBS in a humid chamber for 30 min at 37˚C. The samples were next incubated with primary antibody diluted in blocking buffer for 45 min −2 hr in the humid chamber at 37˚C. The samples were then washed 3 times in 1X PBS/0.2% Tween 20 for 3 min each while shaking. After a 5 min incubation in blocking buffer at 37˚C in the humid chamber, the samples were incubated in blocking buffer containing fluorescently-conjugated secondary antibody for 30 min in the humid chamber at 37˚C, followed by three washes in PBS/0.2% Tween 20 while shaking for 3 min each. For samples undergoing only IF staining, DAPI was added to the third wash at a 1:250,000 dilution. Coverslips were then mounted on slides in Vectashield (Vector Labs, #H-1000). For samples undergoing IF combined with RNA FISH, the samples were processed for RNA FISH following the third wash. Antibody information is described in *Supplementary file 5*.

## RNA fluorescence *in situ* hybridization (RNA FISH)

RNA FISH with double-stranded and strand-specific probes was performed as previously described (*Gayen et al., 2015*; *Hinten et al., 2016*; *Kalantry et al., 2009*). The *Rnf12* dsRNA FISH probe was made by random-priming using BioPrime DNA Labeling System (Invitrogen, #18094011) and labeled with Cy3-dCTP (GE Healthcare, #PA53021) using a previously described fosmid template (*Kalantry et al., 2009*). Strand-specific *Xist* probes were generated from templates as described

(*Maclary et al., 2014*; *Sarkar et al., 2015*). Probes were labeled with Fluorescein-12-UTP (Roche, #11427857910) or Cy5-CTP (GE Healthcare, #25801087). Labeled probes from multiple templates were precipitated in a 0.5M ammonium acetate solution (Sigma, #09691) along with 300 µg of yeast tRNA (Invitrogen, #15401–029) and 150 µg of sheared, boiled salmon sperm DNA (Invitrogen, #15632–011). The solution was then spun at 15,000 rpm for 20 min at 4°C. The pellet was washed consecutively with 70% ethanol and 100% ethanol while spinning at 15,000 rpm at room temperature. The pellet was dried and resuspended in deionized formamide (VWR, #97062–010). The probe was denatured by incubating at 90°C for 10 min followed by an immediate 5 min incubation on ice. A 2X hybridization solution consisting of 4X SSC and 20% Dextran sulfate (Millipore, #S4030) was added to the denatured solution. All probes were stored in the dark at −20°C until use.

Following IF, embryos mounted on coverslips were dehydrated through 2 min incubations in 70%, 85%, 95%, and 100% ethanol solutions and subsequently air-dried. The coverslips were then hybridized to the probe overnight in a humid chamber at 37°C. The samples were then washed 3 times for 7 min each at 37°C with 2X SSC/50% formamide, 2X SSC, and 1X SSC. A 1:250,000 dilution of DAPI (Invitrogen, #D21490) was added to the third 2X SSC wash. Coverslips were then mounted on slides in Vectashield (Vector Labs, #H-1000).

## Allele-specific Xist RNA FISH

Allele-specific Xist RNA FISH probes were generated as described (*Levesque et al., 2013*). Briefly, a panel of short oligonucleotide probes were designed to uniquely detect either the *M. musculus* or the *M. molossinus* alleles of *Xist* (*Supplementary file 5*). Five probes were designed for each *Xist* allele. Each probe overlapped a single nucleotide polymorphism (SNP) that differs between the two strains, with the SNP located at the fifth base pair position from the 5' end. The same panel of five SNPs was used for both sets of allele-specific probes. The 3' end of each oligonucleotide probe is fluorescently tagged using Quasar dyes (Biosearch technologies). *M. musculus*-specific oligos were labeled with Quasar 570 and *M. molossinus*-specific oligos labeled with Quasar 670. In addition to labeled SNP-overlapping oligonucleotides, a panel of 5 'mask' oligonucleotides were also synthesized. These 'mask' probes are complimentary to the 3' end of the labeled allele-specific probes and will hybridize to the allele-specific oligonucleotides, leaving only 9–10 base pairs of sequence surrounding the polymorphic site available to initially hybridize to the target Xist RNA. Since this region of complementarity is short, the presence of a single nucleotide polymorphism is sufficient to destabilize the hybridization with the alternate allele. Sequences of detection and mask probes are listed in *Supplementary file 5*. Allele-specific Xist RNA FISH probes were combined with a strand-specific Xist RNA probe, labeled with Fluorescein-12-UTP (Roche, #11427857910), which served as a guide probe that hybridizes to Xist RNA generated from both *Xist* alleles and ensured the fidelity of the allele-specific probes in detecting the cognate Xist RNA molecules. The guide Xist RNA probe was first ethanol precipitated as previously described, then resuspended in hybridization buffer containing 10% dextran sulfate, 2X saline-sodium citrate (SSC) and 10% formamide. The precipitated guide RNA probe was then mixed with the *M. musculus* and *M. molossinus* detection probes, to a final concentration of 5 nM per allele-specific oligo, and 10 nM mask probe, yielding a 1:1 mask:detection oligonucleotide ratio. Coverslips were hybridized to the combined probe overnight in a humid chamber at 37°C. After overnight hybridization, samples were washed twice in 2X SSC with 10% formamide at 37°C for 30 min, followed by one wash in 2X SSC for five min at room temperature. A 1:250,000 dilution of DAPI (Invitrogen, #D21490) was added to the second 2X SSC with 10% formamide wash. Coverslips were then mounted on slides in Vectashield (Vector Labs, #H-1000).

## Microscopy

Stained samples were imaged using a Nikon Eclipse TiE inverted microscope with a Photometrics CCD camera. The images were deconvolved and uniformly processed using NIS-Elements software. For four color images (blue, green, red, and white), the far-red spectrum was employed for the fourth color (AlexaFluor 647 secondary antibody and Cy5-UTP labelled riboprobes for RNA FISH). Additional antibody information is outlined in *Supplementary file 5*.

EED and H3K27me3 IF intensity quantifications were performed using the '3D Measurement; 3D thresholding, 3D viewing and voxel based measurements' Nikon Elements software package (Nikon Instruments, 77010582). Individual nuclei were marked by creating a binary image, using the

'Threshold' function, over the DAPI stain of the nuclei. Each nucleus was designated as a Region of Interest (ROI) by converting the binary image to an ROI. An additional polygonal ROI was manually created over a non-nuclear region, which was thensubtracted from the nuclear fluorescence intensity. For each channel, average intensity of each nucleus was taken as the intensity measurements from individual ROIs. These intensity values of individual nuclei of an embryo were then averaged to get the average intensity per embryo. Embryos with 2–3 cells were categorized as being at the ~2-cell stage in development. The ~4-cell stage encompassed embryos with 4–5 cells. Embryos with 6–10 cells were classified as being at the ~8-cell stage in development, and the ~16-cell stage encompassed embryos with 14–19 cells. To preserve IF intensities, the images of embryos were not deconvolved. Intensity data for individual nuclei is presented in *Figure 2—source data 1*.

The Threshold function of the software cannot always distinguish between two nuclei that are overlapping. Similarly, if a single nucleus is an odd shape, it may be counted as multiple nuclei by the software. Some embryos therefore had different numbers of nuclei measured than the number of cells in the embryo. If the number of cells in an embryo differs from the number of nuclei listed, the actual number of cells is indicated in parenthesis next to the embryo label in *Figure 2—source data 1*.

## RNA-Seq sample preparation

mRNA was isolated from whole embryos using the Dynabeads mRNA DIRECT Kit (Thermo Fisher, # 610.11) according to the manufacturer's instructions. E3.5 embryos of similar sizes of all genotypes were used for RNA-Seq. *Eed*^fl/- and *Eed*^-/- embryos were genotyped by *Eed* RT-PCR and all embryo genotypes were confirmed by quantifying the relative expression of the floxed *Eed* exon seven to the sample's number of mapped reads (*Figure 3—figure supplement 1* and *Figure 4—figure supplement 1*). Samples were submitted to the University of Michigan DNA Sequencing Core for strand-specific library preparation using the Takara SMARTer Seq V4 stranded low input kit (Takara, #634889). All libraries were sequenced on the Illumina HiSeq2000 or HiSeq4000 platforms to generate 50 bp paired-end reads.

## Mapping of RNA-Seq data

Quality control analysis of the RNA-Seq data was conducted using FastQC. SNP data from whole-genome sequencing of the 129/S1 (*M. musculus*) and JF1/Ms (*M. molossinus*) mouse strains were substituted into the mm9 mouse reference genome build (C57BL/6 J) using VCFtools to generate in silico 129/S1 and JF1/Ms reference genomes (*Keane et al., 2011*; *Maclary et al., 2017*; *Takada et al., 2013*; *Yalcin et al., 2011*). Sequencing reads were separately mapped to each of the two in silico genomes using STAR (*Dobin et al., 2013*), allowing 0 mismatches in mapped reads to ensure allele-specific mapping of SNP-containing reads to only one strain-specific genome. STAR was selected for read mapping, in part due to the improved ability to handle structural variability and indels, with the goal of reducing mapping bias to the genome most similar to the reference genome. STAR is a spliced aligner capable of detecting structural variations and is able to handle small insertions and deletions during read mapping. STAR additionally permits soft-clipping of reads during mapping, trimming the ends of long reads that cannot be perfectly mapped. This function would permit clipping of reads that end near indels, thus preserving mappability at SNPs near indels.

Prior work showed that the variability due to mapping bias between the 129/S1 and JF1/Ms genomes is minimal in our RNA-Seq analysis pipeline (*Maclary et al., 2017*). Although small biases may affect allelic mapping at a subset of SNP sites within a gene, the effect is mitigated since most genes contain multiple SNPs (*Figure 3—figure supplement 1*).

## Allele-specific analysis of RNA-Seq data

For allelic expression analysis, only RNA-Seq reads overlapping known SNP sites that differ between the 129/S1 and JF1/Ms genomes were retained. All multi-mapping reads were excluded from the allele-specific analysis. For each SNP site, reads mapping to the 129/S1 and JF1/Ms X chromosomes were counted and the proportion of reads from each X chromosome identified. Allelic expression was calculated individually for each SNP site; for genes containing multiple SNPs, the paternal-X

percentage for all SNPs was averaged to calculate gene-level allelic expression. All SNP sites with at least 10 SNP-overlapping reads were retained. Genes containing at least one SNP site with at least 10 SNP-overlapping reads were retained for further analysis and are referred to in the text as informative. In X-linked genes, the SNP frequency is ~1 SNP/250 bp in transcribed RNAs (*Keane et al., 2011*; *Maclary et al., 2017*; *Takada et al., 2013*; *Yalcin et al., 2011*).

### RNA-Seq expression analysis

To calculate expression from the maternal vs. paternal X-chromosomes, all reads were first merged into a single alignment file and the number of reads per RefSeq annotated gene was counted using HTSeq. To calculate the percentage of expression arising from the paternal X-chromosome, the total read counts from HTSeq were normalized by number of mapped reads. Then, the normalized number of mapped reads for each gene was multiplied by the proportion of SNP-containing reads mapping to the paternal X-chromosome. This analysis was done in R using the following formula:

$$\left\{ \text{total reads} \times \left( \frac{\text{paternal reads}}{\text{maternal reads} + \text{paternal reads}} \right) \right\}$$

### Analysis of human and mouse oocyte RNA-Seq data

For analysis of publicly available oocyte RNA-Seq data, raw Fastq files were obtained from the NCBI Sequence Read Archive. Quality control analysis was conducted using FastQC. Reads were aligned to the mm9 (mouse) or hg19 (human) reference genome using STAR (*Dobin et al., 2013*) and counted using FeatureCounts (*Liao et al., 2014*). BioProject and Run numbers for samples analyzed are listed here.

| Human oocyte RNA-Seq | | Mouse oocyte RNA-Seq | |
|---|---|---|---|
| **BioProject ID** | **Run number** | **BioProject ID** | **Run number** |
| PRJNA146903 | SRR351336 | PRJDB21 | DRR001701 |
| PRJNA146903 | SRR351337 | PRJDB21 | DRR001702 |
| PRJEB8994 | ERR841204 | PRJNA154207 | SRR385627 |

### Statistical analysis and plots

Welch's two-sample T-tests were used to test for significant differences between the means of Pyrosequencing and RNA-Seq allelic expression data. This test was chosen due to the unequal variance and sample sizes between different genotype groups. In the RNA-Seq allelic expression significance tests, the average percent paternal expression of all informative X-linked genes was calculated for each sample. The total paternal expression value for each genotype group was obtained by calculating the mean of the informative percent paternal values for all samples in that genotype group. A two-tailed Student's T-test was used to determine the significance of RNA FISH and IF data. All barplots and heatmaps were made using the ggplot and Pheatmaps R packages, respectively. Dotplots were made using Python's Seaborn package. Only genes that were informative in all samples were included in the heatmaps.

## Acknowledgements

We thank Paul Ginart and Arjun Raj for help with designing of the allele-specific Xist RNA FISH assay. We also thank Milan Samanta for isolating a subset of the mouse embryos; Emily Maclary for establishing the allele-specific RNA-Seq pipeline used in the study and initiating the RNA-Seq analysis of embryos. We thank Greggory Myers for critically evaluating the manuscript. We acknowledge the services of the University of Michigan Sequencing Core Facility, supported in part by the University of Michigan Comprehensive Cancer Center; and, the University of Michigan Transgenic Animal Model Core Facility.

# Additional information

## Funding

| Funder | Grant reference number | Author |
|---|---|---|
| National Institute of General Medical Sciences | T32GM07544 | Marissa Cloutier Megan Trotter |
| Howard Hughes Medical Institute | International Scholar | Wei Xie |
| NIH Office of the Director | DP2-OD-008646 | Sundeep Kalantry |
| National Institutes of Health | R01GM124571 | Sundeep Kalantry |
| National Institutes of Health | R01HD095463 | Sundeep Kalantry |
| March of Dimes Foundation | 5-FY12-119 | Sundeep Kalantry |
| University of Michigan | Endowment for the Basic Sciences | Sundeep Kalantry |
| NationalNatural Science Foundation of China | 31725018 | Wei Xie |
| Beijing Municipal Science & Technology Commission | Z181100001318006 | Wei Xie |

The funders had no role in study design, data collection and interpretation, or the decision to submit the work for publication.

## Author contributions

Clair Harris, Conceptualization, Data curation, Formal analysis, Validation, Investigation, Visualization, Methodology, Writing—original draft, Writing—review and editing; Marissa Cloutier, Conceptualization, Data curation, Formal analysis, Funding acquisition, Validation, Investigation, Visualization, Methodology, Writing—original draft, Writing—review and editing; Megan Trotter, Data curation, Formal analysis, Funding acquisition, Validation, Investigation, Visualization, Methodology, Writing—original draft, Writing—review and editing; Michael Hinten, Formal analysis, Investigation, Methodology, Writing—review and editing; Srimonta Gayen, Investigation, Writing—review and editing; Zhenhai Du, Wei Xie, Resources, Data curation, Formal analysis; Sundeep Kalantry, Conceptualization, Data curation, Formal analysis, Supervision, Funding acquisition, Validation, Investigation, Visualization, Methodology, Writing—original draft, Project administration, Writing—review and editing

## Author ORCIDs

Clair Harris (iD) http://orcid.org/0000-0003-3609-7417
Marissa Cloutier (iD) http://orcid.org/0000-0001-7078-542X
Megan Trotter (iD) http://orcid.org/0000-0001-5895-2926
Sundeep Kalantry (iD) http://orcid.org/0000-0003-0763-8050

## Ethics

Animal experimentation: This study was performed in strict accordance with the recommendations in the Guide for the Care and Use of Laboratory Animals of the National Institutes of Health. All animals were handled according to protocols approved by the University Committee on Use and Care of Animals (UCUCA) at the University of Michigan (protocol #s PRO6455 and PRO8425).

## Decision letter and Author response

Decision letter https://doi.org/10.7554/eLife.44258.032
Author response https://doi.org/10.7554/eLife.44258.033

## Additional files

### Supplementary files

• Supplementary file 1. Pairwise analysis of IF intensity data in *Figure 2*. Statistical comparisons of EED and H3K27me3 IF intensities between all genotypes at each embryonic stage analyzed in *Figure 2* (Two-tailed Student's T-test).
DOI: https://doi.org/10.7554/eLife.44258.017

• Supplementary file 2. Percentage of total X-linked gene expression from the paternal X-chromosome by RNA-Seq. Analysis of each informative X-linked gene (>10 reads/SNP).
DOI: https://doi.org/10.7554/eLife.44258.018

• Supplementary file 3. Pairwise analysis of RNA-Seq data. Statistical comparisons of percent allelic expression from the paternal X-chromosome and normalized expression from the paternal- or maternal-X of all genotypes.
DOI: https://doi.org/10.7554/eLife.44258.019

• Supplementary file 4. Pairwise analysis of Pyrosequencing data. Statistical comparisons of Pyrosequencing data of *Xist*, *Rnf12*, *Atrx*, and *Pgk1* RNAs in embryos of all genotypes.
DOI: https://doi.org/10.7554/eLife.44258.020

• Supplementary file 5. Primer sequences and antibody information. Primer sequences for Pyrosequencing, RT-PCR, genomic PCR, and allele-specific RNA FISH probes; and, antibody information.
DOI: https://doi.org/10.7554/eLife.44258.021

• Transparent reporting form
DOI: https://doi.org/10.7554/eLife.44258.022

### Data availability

Sequencing data have been deposited in GEO under accession number GSE123173. The analyzed sequencing data are also included as Supplementary File 3.

The following dataset was generated:

| Author(s) | Year | Dataset title | Dataset URL | Database and Identifier |
|---|---|---|---|---|
| Harris C | 2019 | Conversion of Random X-inactivation to Imprinted X-inactivation by Maternal PRC2 | https://www.ncbi.nlm.nih.gov/geo/query/acc.cgi?acc=GSE123173 | NCBI Gene Expression Omnibus, GSE123173 |

The following previously published datasets were used:

| Author(s) | Year | Dataset title | Dataset URL | Database and Identifier |
|---|---|---|---|---|
| Reich A, Klatsky P, Carson S, Wessel G | 2011 | The transcriptome of a human polar body accurately reflects its sibling oocyte | https://www.ncbi.nlm.nih.gov/pubmed/21953461 | NCBI BioProject, PRJNA146903 |
| Karolinska Institutet | 2015 | Gene expression during the first three days of human development | https://www.ncbi.nlm.nih.gov/bioproject/PRJEB8994 | NCBI BioProject, PRJEB8994 |
| Kobayashi H, Sakurai T, Imai M | 2012 | Contribution of intragenic DNA methylation in mouse gametic DNA methylomes to establish oocyte-specific heritable marks | https://www.ncbi.nlm.nih.gov/bioproject/PRJDB21 | NCBI BioProject, PRJDB21 |

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
