## [Decision Letter]

Thank you for submitting your article "Conversion of Random X-inactivation to Imprinted X-inactivation by Maternal PRC2" for consideration by *eLife*. Your article has been reviewed by three peer reviewers, and the evaluation has been overseen by a Reviewing Editor and Jessica Tyler as the Senior Editor. The following individuals involved in review of your submission have agreed to reveal their identity: Mauro Calabrese (Reviewer #1); Bernhard Payer (Reviewer #3).

The reviewers have discussed the reviews with one another and the Reviewing Editor has drafted this decision to help you prepare a revised submission.

Summary:

You have investigated the role of the Polycomb repressive complex 2 (PRC2) member EED for imprinted X-chromosome inactivation in mice. You find that mostly maternal expression during oogenesis is important and that conditional deletion of EED in oocytes leads to a shift from imprinted to random X-inactivation. You nicely show that oocyte ablation of *Eed* compromises the initiation of imprinted X chromosome inactivation. In contrast zygotic deletion of *Eed* does not exhibit this failure. However, maternal and zygotic deletion of *Eed* cause a more severe defect. Using allele-specific FISH with F1 hybrid embryos, you demonstrate that maternal null female embryos subsequently stochastically silenced Xist from one of the two X chromosomes, leading to random X inactivation. These are important contributions to our understanding of X inactivation.

Essential revisions:

Essential Experimental Revisions

1) All three reviewers agree that the authors must test embryo genotypes in their experiments for *Eed* deletion. Specific comments from the reviewers are:

"Figure 1B shows embryos derived from a cross between from a cross of *Eed^+/-^* females with *Eed^fl/-^*;Prm-Cre males. They do not genotype the embryos and as a result infer which embryos are null for Eed and which are heterozygous. Likewise in Figure 2D, E the genotype of the blastocysts was inferred. "

"Figure 4C shows the number of *Eed^m-/-^* embryos that can live to term compared to *Eed^fl/fl^*. Did the investigators genotype the embryos to be assured that the Cre really did excise *Eed* in the oocyte? Also, can the authors statistically analyze the data to determine if surviving females are in fact overrepresentated relative to males?"

"Figure 4—figure supplement 1E depicts allele-specific H3K27me3 ChIP-seq at the Xist locus of MII oocyte, sperm, PN5 zygote, 8-cell embryo and inner cell mass from Zhang et al., 2017. What is the genotype of these embryos?"

2) The reviewers also expressed concern about the levels of Eed protein after deletion, since deletions can occur at different time points after Cre expression. For this the authors should test levels of *Eed* in their embryos by immunofluorescence. Specific comments from the reviewers are:

"Also, because the Cre may delete later, the authors suggest that some embryos have intermediate levels of *Eed*. Given the uncertainty of the actual level *Eed* in the embryos, this experiment is not really convincing".

"It is not clear, to which degree the conditional EED knockout allele of this study completely removes EED from early embryos and when zygotic EED expression would kick in. The authors should provide therefore EED/H3K27me3 antibody double-stainings for *Eed^fl/fl^, Eed^-/-^, Eed^m-/-^* and *Eed^mz-/-^* embryos throughout preimplantation development (zygote, 2-cell, 4-cell, 8-cell, 16-cell). This will provide information, how leaky/complete the *Zp3-Cre* and *Prm-Cre* deletions might be, which potentially could explain the variability of the phenotype between embryos. Furthermore, this will tell, how long the maternal protein is stable (in *Eed^-/-^*) and when the zygotic expression (in *Eed^m-/-^*) will begin, thereby better defining the time-window, when the maternal and zygotic EED protein is functional and giving more insight for interpreting the observed phenotypes. Finally it will also show the consequence and kinetics for H3K27me3 accumulation".

Essential editorial revisions

1) "First line of Abstract: "Imprinted X-inactivation silences genes exclusively on the paternally-inherited chromosome and is a paradigm of transgenerational epigenetic inheritance in mammals." This is untrue-X inactivation is not a paradigm of "transgenerational" epigenetic inheritance. Maybe epigenetic inheritance. Please remove "transgenerational." This is also used in the first paragraph of the Introduction and in the Discussion."

2) "Discussion, sixth paragraph: In this section the authors discuss their data in respect to the recent similar study by Inoue et al., 2018. The authors claim that the other study drew their conclusions mainly from postimplantation stages but does not provide information about the initiation or establishment of the imprinting loss of this X-inactivation state. This is simply not true, as the paper by Inoue et al. does indeed also analyse data from morula and blastocyst stage embryos, similar to the current study. Also does the study by Inoue observe the phenotype in maternal-only Eed mutants, thereby clearly being able to conclude that maternal, but not zygotic EED is responsible for the imprint on X-inactivation. The authors of the current study should therefore tone down their wording of this section and give proper credit to the findings from Inoue et al., which are in agreement and confirmed by the current study."

---

## [Author Response]

Essential revisions:Essential Experimental Revisions1) All three reviewers agree that the authors must test embryo genotypes in their experiments for Eed deletion. Specific comments from the reviewers are:"Figure 1B shows embryos derived from a cross between from a cross of Eed^+/-^ females with Eed^fl/-^;Prm-Cre males. They do not genotype the embryos and as a result infer which embryos are null for Eed and which are heterozygous. Likewise in Figure 2D, E the genotype of the blastocysts was inferred. "

These are valid points. To genotype the embryos for *Eed* in Figure 1B, we assayed enrichment of the EED protein and the PRC2-catalyzed H3K27me3 mark, for which EED is essential, on the inactive-X, which we marked by Xist RNA FISH. The inactive-X enrichment of EED and H3K27me3 provides a robust assay to determine the presence or absence of EED in the embryo. In wild-type (WT) embryos, EED and/or H3K27me3 are enriched on the Xist RNA-coated inactive-X in >70% of the nuclei (Figure 1A).

We therefore classified embryos in Figure 1B devoid of any nuclei with EED/H3K27me3 enrichment on the Xist RNA-coated X-chromosome as genotypically *Eed^-/-^*(9 of 41 embryos). Conversely, we classified embryos in Figure 1B as *Eed^+/-^* if they had statistically indistinguishable percentage of their nuclei with coincident enrichment of EED and/or H3K27me3 on the Xist RNA-coated X-chromosome as in the WT embryos in Figure 1A (9 of 41 embryos as *Eed^+/-^* in Figure 1B). Of note, in the original submission we did not statistically compare this class of embryos with the WT embryos in Figure 1A. Doing so now allows us to confidently conclude that these embryos are *Eed^+/-^*.

The remaining 23 embryos in Figure 1B had some nuclei with EED/H3K27me3 enrichment on the Xist RNA coated X-chromosome. But, the frequency of such enrichment was significantly less than the WT embryos in Figure 1A and also significantly less than the *Eed^+/-^* embryos in Figure 1B. This class of embryos could either be *Eed^+/-^* or *Eed^-/-^* embryos that have not yet fully depleted maternally-inherited EED protein. Or, they could be *Eed^+/-^* embryos that have not yet robustly expressed zygotic EED. The identification of this intermediate class of embryos suggests that the expression of Eed transitions from maternal to zygotic at or slightly before the blastocysts stage. The objective of Figure 1B was to capture this intermediate class of embryos, which would inform how long maternal EED mRNA/protein persists during early embryonic development.

As an independent test of the genotype and sex of embryos derived from the cross of *Eed^+/-^* females with *Eed^fl/-^;Prm-Cre* males in Figure 1B, we performed PCR genotyping of embryos generated from the cross and present these data in Figure 1C. Embryos from 12 litters showed the expected distribution of *Eed^+/-^* and *Eed^-/-^* male and female embryos. Together, the results in Figure 1B-C suggest that genotypically null *Eed^-/-^*embryos inherit oocyte-derived maternal EED protein and that the expression of EED transitions from maternal to zygotic prior to the blastocyst stage.

With regards to Figure 3D and 3E (previously Figure 2D and 2E), we ascertained the Eed genotype of these embryos by assaying for H3K27me3 enrichment on the Xist RNA-coated X-chromosome. Only embryos in which >95% of the nuclei were devoid of H3K27me3 accumulation on the Xist RNA-coated X-chromosome were classified as *Eed^-/-^*. Much previous work by us and by others has demonstrated that EED is essential for the catalysis of H3K27me3 (see, for example, Montgomery et al., 2005; Kalantry et al., 2006; Kalantry and Magnuson, 2006). Thus, assaying H3K27me3 enrichment on the inactive-X is an acceptable proxy for *Eed* genotyping. In some respects, H3K27me3 accumulation may in fact be a preferred genotyping method for *Eed*, since H3K27me3 is the functional readout of the presence of EED protein."Figure 4C shows the number of Eedm^-/-^ embryos that can live to term compared to Eed^fl/fl^. Did the investigators genotype the embryos to be assured that the Cre really did excise Eed in the oocyte? Also, can the authors statistically analyze the data to determine if surviving females are in fact overrepresentated relative to males?"

Yes, we genotyped the liveborn mice from the crosses and found that *Eed^m-/-^* animals that are obtained from a cross of *Eed^fl/fl^;Zp3-Cre* females with wild-type males are all genotypically *Eed^-/+^*. We now include this information in the table in Figure 5C. We also include statistical analysis in the revised manuscript to test whether the over-representation of males is statistically significant. We apologize for not including this information in the original submission.

"Figure 4—figure supplement 1E depicts allele-specific H3K27me3 ChIP-seq at the Xist locus of MII oocyte, sperm, PN5 zygote, 8-cell embryo and inner cell mass from Zhang et al., 2017. What is the genotype of these embryos? "

We again apologize for not including this information earlier. These are profiles of wild-type oocytes and embryos. We now include this information in the figure and the figure legend.

2) The reviewers also expressed concern about the levels of Eed protein after deletion, since deletions can occur at different time points after Cre expression. For this the authors should test levels of Eed in their embryos by immunofluorescence. Specific comments from the reviewers are:"Also, because the Cre may delete later, the authors suggest that some embryos have intermediate levels of Eed. Given the uncertainty of the actual level Eed in the embryos, this experiment is not really convincing".

We assume this comment is in reference to the blastocyst embryos shown in Fiure 1B, which are derived from a cross of *Eed^+/-^*females with *Eed^fl/fl^;Prm-Cre* males. A subset of these embryos show variable levels of EED and/or H3K27me3 enrichment on the inactive-X. *Prm-Cre* is active during spermatogenesis and catalyzes the deletion of the loxp flanked (‘fl’) alleles in the mature sperm (O'Gorman et al., 1997. If expression of the *Prm-Cre* transgene is fully penetrant, the embryos from the above cross should genotypically either be *Eed^+/-^* or *Eed^-/-^*. The reviewer’s comment led us to experimentally assess the efficiency of Prm-Cre-mediated deletion of the *Eed^fl^*allele. We genotyped liveborn animals from a cross of *Eed^fl/fl^*or WT females with *Eed^fl/fl^;Prm-Cre* or *Eed^fl/-^;Prm-Cre* males. Through this analysis, we found that the *Prm-Cre* transgene excises the *Eed^fl^*allele very efficiently: 89% and 91% of the derived animals inherit a deleted *Eed* allele from *Eed^fl/fl^;Prm-Cre* or *Eed^fl/-^;Prm-Cre* males, respectively. We include these data in Figure 1—figure supplement 1B.

To test the penetrance of the Zp3-Cre transgene (Lewandoski et al., 1997), we genotyped liveborn animals born from *Eed^fl/fl^;Zp3-Cre* females crossed to WT males. 100% of the mice derived from this cross inherited a deleted *Eed* allele. Therefore, the *Zp3-Cre* transgene is 100% efficient in deleting *Eed^fl^* allele. We now provide this information in the table in Figure 5C.

As a further test of CRE efficiency, we compared *Eed* expression in the RNA-Seq data from *Eed^fl/-^, Eed^-/-^, Eed^m-/-^*, and *Eed^mz-/-^* blastocyst embryos (Figure 3—figure supplement 1A and Figure 4—figure supplement 1A). *Eed^mz-/-^*embryos displayed negligible expression of *Eed*, again attesting to the highly penetrant excision of *Eed^fl^* alleles in the oocyte and the sperm.

In sum, the highly penetrant excision of *Eed^fl^* alleles suggests that the variation in EED and/or H3K27me3 inactive-X enrichment is unlikely due to the inefficient excision of the *Eed^fl^*allele by *Prm-Cre* or Zp3-Cre.

"It is not clear, to which degree the conditional EED knockout allele of this study completely removes EED from early embryos and when zygotic EED expression would kick in. The authors should provide therefore EED/H3K27me3 antibody double-stainings for Eed^fl/fl^, Eed^-/-^, Eed^m-/-^ and Eed^mz-/-^ embryos throughout preimplantation development (zygote, 2-cell, 4-cell, 8-cell, 16-cell). This will provide information, how leaky/complete the Zp3-Cre and Prm-Cre deletions might be, which potentially could explain the variability of the phenotype between embryos. Furthermore, this will tell, how long the maternal protein is stable (in Eed^-/-^) and when the zygotic expression (in Eed^m-/-^) will begin, thereby better defining the time-window, when the maternal and zygotic EED protein is functional and giving more insight for interpreting the observed phenotypes. Finally it will also show the consequence and kinetics for H3K27me3 accumulation".

As described above, *Prm-Cre* and *Zp3-Cre* are both highly efficient at excising the *Eed^fl^*allele. These breeding data, however, do not inform the timing of the Cre-mediated deletion of the *Eed^fl^*allele, especially by *Zp3-Cre* during oogenesis. As the reviewer suggests, if the *Zp3-Cre* mediated deletion is late enough during oogenesis, then some maternal Eed mRNA/protein may be transmitted to the embryo. This residual *Eed* product could lead to variability in imprinted X-inactivation defects in *Eed^m-/-^* and *Eed^mz-/-^* embryos.

Importantly, though, we do not observe such variation in very early *Eed^m-/-^* and *Eed^mz-/-^* embryos when X-inactivation normally begins. Just after fertilization, in 3-cell *Eed^m-/-^* female embryos and in 3-5 cell *Eed^m-/-^* male embryos, all nuclei display ectopic Xist RNA induction from and coating of the maternal X-chromosome (Figure 7A-B [previously Figure 6A-B]). This uniform ectopic Xist RNA induction from the maternal-X, therefore, indicates that *Zp3-Cre* mediated deletion of the *Eed^fl^* alleles occurs early enough in oogenesis to preclude transmission to the embryo of oocyte-generated Eed mRNA/protein. This transmission, through catalysis of H3K27me3, would be responsible for preventing Xist RNA induction from the maternal X-chromosome.

Nevertheless, we agree with the reviewers’ suggestion to test the kinetics of the maternal:zygotic expression transition of EED at the earlier stages of embryogenesis. We therefore generated 2-, 4-, 8-, and 16-cell embryos of the five genotypes, *Eed^fl/fl^, Eed^fl/-^, Eed^-/-^, Eed^m-/-^*, and *Eed^mz-/-^*, and immunofluorescently stained them for EED and H3K27me3. We then quantified the relative levels of EED and H3K27me3 in the different genotypes using an automated function of the Nikon NIS-Elements software. We provide these new data as Figure 2. In brief, we found that maternal expression of EED persists at least until the 16-cell embryo stage and zygotic expression begins at about the 4-cell stage. Please see the subsection “EED and H3K27me3 Enrichment on the Inactive-X in *Eed^-/-^* Embryos” for a detailed description of these data. Of note, we did not generate staining data from zygotes because to generate early preimplantation embryos we superovulated females, which often yield unfertilized eggs. It is sometimes difficult to distinguish unfertilized eggs from zygotes.

Essential editorial revisions1) "First line of Abstract: "Imprinted X-inactivation silences genes exclusively on the paternally-inherited chromosome and is a paradigm of transgenerational epigenetic inheritance in mammals." This is untrue-X inactivation is not a paradigm of "transgenerational" epigenetic inheritance. Maybe epigenetic inheritance. Please remove "transgenerational." This is also used in the first paragraph of the Introduction and in the Discussion."

We apologize for the confusion here. But, genomic imprinting is often cited as an example of transgenerational epigenetic inheritance, because of the parent-of-origin-dependent expression of two homologous alleles in the embryo (see, for example, van Otterdijk and Michels, 2016; Grossniklaus et al., Nature Reviews Genetics [2013]; and, Ferguson-Smith and Bourc’his, 2018).

Like autosomally imprinted genes, imprinted X-inactivation also fits the definition of transgenerational epigenetic inheritance, since the paternal but not maternal X-chromosome is subject to transcriptional inactivation despite the equivalent sequence of the two X-chromosomes and their shared residence in the nucleoplasm. This pattern of inactivation reflects an epigenetic difference in the maternal vs. the paternal X-chromosome that is established in the germline of the parents which then influences gene expression in the offspring. That the parents can influence gene expression in the offspring makes imprinted X-inactivation transgenerational. Our work (and the work by Inoue et al.) implicates the H3K27me3 mark that is catalyzed by maternal PRC2 as the epigenetic imprint that underlies imprinted X-inactivation.

2) "Discussion, sixth paragraph: In this section the authors discuss their data in respect to the recent similar study by Inoue et al., 2018. The authors claim that the other study drew their conclusions mainly from postimplantation stages but does not provide information about the initiation or establishment of the imprinting loss of this X-inactivation state. This is simply not true, as the paper by Inoue et al. does indeed also analyse data from morula and blastocyst stage embryos, similar to the current study. Also does the study by Inoue observe the phenotype in maternal-only Eed mutants, thereby clearly being able to conclude that maternal, but not zygotic EED is responsible for the imprint on X-inactivation. The authors of the current study should therefore tone down their wording of this section and give proper credit to the findings from Inoue et al., which are in agreement and confirmed by the current study."

To address this point, we again reviewed the Inoue et al. manuscript in detail. Below, we discuss the reasoning behind the discussion of Inoue et al. in our submission.

Inoue et al. perform Xist RNA FISH to show that *Eed^m-/-^* morula-stage, E3.5 blastocysts, and E4.0 blastocysts display ectopic Xist RNA induction. In both male and female *Eed^m-/-^* preimplantation embryos, the frequency of nuclei with ectopic Xist RNA coating decreases as the embryos develop. From these data, however, Inoue et al. cannot know which *Xist* allele is being silenced. In male *Eed^m-/-^* embryos, *Xist* is ectopically expressed from their maternally-inherited X-chromosome (XY males’ sole X-chromosome is inherited from their mothers). Since *Eed^m-/-^* males extinguish ectopic Xist RNA expression from their maternal-X, it’s very possible that *Eed^m-/-^* females also silence ectopic Xist RNA expression from their maternal-X. The silencing of ectopic Xist RNA expression from the maternal-X would restore imprinted X-inactivation of the paternal X-chromosome in *Eed^m-/-^* female embryos, since only the paternal-X would express Xist. Thus, from the RNA FISH data alone it is not possible to conclude that imprinted X-inactivation is switching to random X-inactivation in *Eed^m-/-^* female embryos.

To test for gene expression across the X-chromosome, Inoue et al. also perform RNA-Seq in both morulas as well as in embryonic day (E)6.5 extraembryonic ectoderm tissues, which normally maintain imprinted X-inactivation. However, at the morula stage the authors pooled male and female embryos and do not analyze female embryos selectively. Nor do the authors test X-linked gene expression in an allele-specific manner. Thus, from the morula-stage RNA-Seq data it is not clear if the ectopic inactivation of the maternal X-chromosome is the cause of defective X-linked gene expression in *Eed^m-/-^* females.

Inoue et al. do perform allele-specific RNA-Seq on extraembryonic tissues from individual *Eed^m-/-^* E6.5 embryos. The E6.5 extra-embryonic tissues normally stably maintain imprinted X-inactivation. Here, the authors observe high variability in allele-specific X-linked gene expression in the four *Eed^m-/-^* samples they analyze. Two out of four *Eed^m-/-^* E6.5 extraembryonic samples exhibit close to equal expression of many genes across the X-chromosome. The other two samples, however, display maternal-X biased gene expression, nearing the pattern observed in control E6.5 extraembryonic tissues. These data compel the authors to conclude that *Eed^m-/-^* female embryos switch their X-inactivation status from imprinted to random.

However, prior work by us found that E6.5 extraembryonic tissues from zygotic *Eed^-/-^* embryos derepress paternal X-linked genes, without the cells switching their X-inactivation status from imprinted to random (Kalantry et al., 2006). The derepression of paternal X-linked genes would result in pattern of X-linked gene expression similar to that Inoue et al. find in their *Eed^m-/-^* samples.

To distinguish amongst the two possibilities – that of extraembryonic cells converting imprinted X-inactivation to random X-inactivation or the lack of stable maintenance of imprinted inactivation of the paternal X-chromosome – requires allele-specific gene expression analysis at the single cell level. These two possibilities pushed us to develop the allele-specific Xist RNA FISH approach that we use to demonstrate that imprinted X-inactivation indeed switches to random X-inactivation as early as the blastocyst stage. To provide direct evidence of imprinted X-inactivation switching to random X-inactivation requires allele-specific gene expression analysis in individual cells and selectively in female embryos.

In sum, we feel that our study offers concrete data for the conclusions that Inoue et al. make. We understand the spirit of the comment and have modulated our language in the discussion of the Inoue et al. manuscript.